# Towards Learning Universal Hyperparameter Optimizers with Transformers

**Yutian Chen**[1], **Xingyou Song**[2], **Chansoo Lee**[2], **Zi Wang**[2], **Qiuyi Zhang**[2],
**David Dohan**[2], **Kazuya Kawakami**[1], **Greg Kochanski**[2],
**Arnaud Doucet**[1], **Marc'aurelio Ranzato**[1], **Sagi Perel**[2], **Nando de Freitas**[1]
[1]Deepmind, [2]Google Research, Brain Team

## Abstract

Meta-learning hyperparameter optimization (HPO) algorithms from prior experiments is a promising approach to improve optimization efficiency over objective functions from a similar distribution. However, existing methods are restricted to learning from experiments sharing the same set of hyperparameters. In this paper, we introduce the OPTFORMER, the first text-based Transformer HPO framework that provides a universal end-to-end interface for jointly learning policy and function prediction when trained on vast tuning data from the wild, such as Google's Vizier database, one of the world's largest HPO datasets. Our extensive experiments demonstrate that the OPTFORMER can simultaneously imitate at least 7 different HPO algorithms, which can be further improved via its function uncertainty estimates. Compared to a Gaussian Process, the OPTFORMER also learns a robust prior distribution for hyperparameter response functions, and can thereby provide more accurate and better calibrated predictions. This work paves the path to future extensions for training a Transformer-based model as a general HPO optimizer.

## 1  Introduction

The emergence of public machine learning data platforms such as OpenML [1] and hyperparameter optimization (HPO) services such as Google Vizier [2], Amazon SageMaker [3] and Microsoft Azure [4] have made large-scale datasets containing hyperparameter evaluations accessible. For our use-case in this paper, Google Vizier is the de-facto HPO service across Google, having optimized some of Google's largest products and research efforts, and contains a collection of valuable tuning data within the last 5 years. While there is growing interest in leveraging such data to meta-learn hyperparameter optimization algorithms [5–8], dealing with large datasets consisting of experimental trials in the wild can be challenging, due to large variations in HPO problems and their associated text metadata (e.g. shown later in Table 1).

Thus, most meta and transfer-learning HPO methods [7–16] consider a restrictive setting where all tasks must share the same set of hyperparameters so that the input data can be represented as fixed-sized vectors. Consequently, such methods only exploit a small portion of the available data to learn priors. This drawback is more severe for large datasets which contain significant amounts of useful information.

To overcome these limitations, we introduce the OPTFORMER, a general hyperparameter optimization framework based on Transformers [17]. Transformers have demonstrated excellent performance in many data tasks, ranging from natural language [18], images [19, 20], biological data [21, 22], code [23, 24], and control [25, 26]. Here, we investigate how to use a Transformer as a universal interface for modelling experimental data and learn HPO algorithms, as given a sufficient amount of data, a Transformer can potentially learn a more complex prior distribution than standard Bayesian Optimization (BO) with Gaussian Processes (GPs), especially as the Transformer possesses certain computational advantages over GPs for large datasets.

---

Code: https://github.com/google-research/optformer. Google AI Blog: https://ai.googleblog.com/2022/08/optformer-towards-universal.html.

36th Conference on Neural Information Processing Systems (NeurIPS 2022).

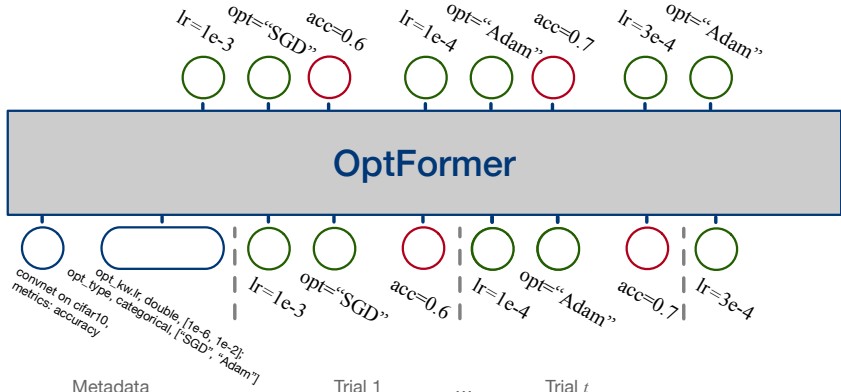

Figure 1: Illustration of the OPTFORMER model over a hyperparameter optimization trajectory. It is trained to predict both hyperparameter suggestions (in green) and response function values (in red).

We introduce a serialization scheme to convert a combination of any metadata and an optimization trajectory into text, represented as a sequence of tokens, and formulate the HPO task as a sequence modeling problem. We adopt a supervised learning approach, by learning to predict parameters and hyperparameter response functions from offline tuning data (See Fig. 1). In order to further improve optimization performance, we augment the model by utilizing its own function prediction during inference (Section 4.3). Extensive experiments on both public and private datasets demonstrate the OPTFORMER's competitive tuning and generalization abilities.

In summary, our contributions are as follows:

- We formulate, to the best of our knowledge, the first meta-learning HPO framework to learn both **policy** and **function priors** from data across different search spaces.
- The OPTFORMER is capable of learning the behaviors of 7 diverse blackbox optimization algorithms relying on a broad class of methods (non-adaptive, evolutionary, and Bayesian).
- Furthermore, the OPTFORMER learns the prior over objective functions and provides both accurate and well calibrated predictions, in many cases significantly surpassing GPs in log-predictive likelihood and expected calibration error (ECE) [27].
- Lastly, OPTFORMER policies augmented with model-based optimization, such as the use of Expected Improvement acquisition functions, are competitive HPO algorithms. To the best of our knowledge, this is the first time Transformers are augmented with acquisition functions for online adaptation.

## 2 Preliminaries

### 2.1 Meta-learning for hyperparameter optimization

HPO aims to find a set of hyperparameters $x$ from search space $\mathcal{X}$ to maximize a model performance metric, $y = f(x)$, often referred to as a response function. Table 1 shows an example of HPO experimental data. Following the HPO nomenclature [2, 28], an experimental study consists of metadata ($m$) and a history of trials ($h$). The metadata contains arbitrary unstructured information, including but not limited to descriptions of the problem, optimization algorithm, names, types and value ranges of hyperparameters. The history after $t$ trials, $h_t = (x_1, y_1, \ldots, x_t, y_t)$, contains a sequence of trials, each of which consists of a parameter suggestion $x$ and function value $y$.

The goal of the meta-learning approach for HPO is to learn the shared knowledge among the objective functions $f$ from a dataset of multiple tuning experiments represented as studies and to obtain an optimal HPO algorithm for new hyperparameter tuning tasks from a similar distribution to those in the dataset.

An HPO algorithm $\pi$ maps the metadata and history to a distribution over hyperparameter suggestions, i.e. $\pi(x_{t+1}|m, h_t)$. Using the terminology of offline RL [29], we refer to the algorithm used to generate the trajectories in a dataset as the behavior policy $\pi_b$.

We primarily consider search spaces $\mathcal{X}$ with a fixed number $D$ of hyperparameters per task, and hence $\boldsymbol{x} = (x^{(1)}, \ldots, x^{(D)})$, with each hyperparameter $x^{(d)}$ being of type `DOUBLE`, `INTEGER`, `DISCRETE`, or `CATEGORICAL` (see Appendix A.1 for details). More complex search spaces can be supported as discussed in Section 7.

## 2.2 Transformer model

The Transformer model is an efficient attention-based neural network architecture for sequence modeling [17]. We adopt the T5 Transformer encoder-decoder architecture [30]. The encoder and decoder each consist of a stack of multi-head self-attention layers which construct pairwise interactions between positions, followed by position-wise feed-forward networks. The encoder converts a sequence of input token representations $m$, to a sequence of continuous embeddings, which is fed to the decoder to generate a sequence of output tokens $h$ one element at a time (see Fig. 1).

## 3 Related work

There has been a rich set of works in meta-learning and transfer learning by modifying specific core components of the BO pipeline, such as the acquisition function or the GP, in order to tackle BO's myopic behavior, or obtaining more information from similar tasks. For instance, approaches include learning new acquisition functions [31], multi-task BO [7–13] and BO for transfer learning using contextual GPs [14–16]. [32] also studies the use of meta-BO for hyperparameter tuning tasks in machine learning. However, all of these works consider a fixed search space.

A more radical meta-learning approach to non-differentiable optimization trains recurrent neural networks (RNNs) as neural optimizers from scratch. [33] first proposed training an RNN with gradient descent to optimize blackbox functions and hyperparameters while [34, 35] train RNNs using reinforcement learning (RL) to solve RL tasks. Unfortunately, prior works are limited to fixed search spaces and only use online generated data, constraining the training objectives to be cheaply computable.

In this work, we wish to overcome the limitations of previous works by exploiting the Transformer architecture. Numerous works have demonstrated Transformers' strong capabilities in flexible symbolic and numerical manipulation. On the symbolic side, Transformers have been shown able to manipulate symbolic mathematical expressions [36–38] and generate code [23, 24]. Furthermore, on the numerical side, Transformers have also been shown able to perform linear algebra computations [39], Bayesian Inference [40], and offline RL [25, 26, 41]. For AutoML specifically, [42] has demonstrated Transformers' and analogous graph neural networks' abilities to use dataset descriptions and metadata to generate classification and data preprocessing pipelines. However, to date, there has been little effort in attacking the full problem of hyperparameter tuning in the blackbox optimization setting. In this paper, the challenging task of learning algorithms from blackbox optimization trajectories can be seen as a significant extension of both symbolic and numerical manipulation. Since the underlying algorithm can be composed of multiple symbolic and mathematical operations with unbounded complexity, the model must infer potentially very complex behavior over long horizons.

## 4 Universal interface and model for hyperparameter optimization

In this section, we provide a universal interface for modeling HPO studies with mixed textual and numerical information as a sequence of discrete tokens. We train our OPTFORMER as a generative

Table 1: Example of a study $(m, \boldsymbol{h})$ with two parameters and two trials. Metadata $m$ appears in blue and history $\boldsymbol{h}$ in purple.

```
"name": "convnet on cifar10",
"metric": "accuracy",
"goal": "MAXIMIZE",
"algorithm": "random_search",
"parameter": {
"name":  "opt_kw.lr",
"type":  "DOUBLE",
"min_value":  1e-6,
"max_value":  1e-2,
"scale_type":  "LOG"
}
"parameter": {
"name":  "opt_type",
"type":  "CATEGORICAL",
"categories":  ["SGD", "Adam"],
}
"trial" {
"parameter":  {
"opt_kw.lr":  0.0021237573,
"opt_type":  "SGD"
}
"metric":  {
"accuracy":  0.69482429,
}}
"trial" {
"parameter":  {
"opt_kw.lr":  0.00038292234,
"opt_type":  "Adam"
}
"metric":  {
"accuracy":  0.71642583
}}
```

Table 2: Serialized study after preprocessing and tokenization. Metadata $m$ appears in blue, normalized and quantized values of $\boldsymbol{x}_t$ in green, and $y_t$ in red.

| After preprocessing | <name>:"convnet on cifar10",<metric>:"accuracy",<goal>:<MAXIMIZE>, <algorithm>:"random_search" &<name>:"opt_kw.lr",<type>:<DOUBLE>,<min_value>:1e-6,<max_value>:1e-2, <scale_type>:<LOG> &<name>:"opt_type",<type>:<CATEGORICAL>,<categories>:["SGD", "Adam"] <831><0>⋆ <0>\| <645><1>⋆ <999> |
|---|---|
| Subwords after tokenization | name : " con v net on ci far 10 ", metric : " acc u racy ", goal : MAXIMIZE , algorithm : " random _ search " & name : " op t _ kw . lr ", type : DOUBLE , min_value : 1 e -6 , max_value : 1 e -2 , scale_type : LOG & name : " op t _ type ", type : CATEGORICAL , categories : [ " SG D ", " A dam " ] 831 0 ⋆ 0 \| 645 1 ⋆ 999 |

model on a given dataset and explain how to use the OPTFORMER's parameter and function prediction abilities to implement an HPO policy.

## 4.1 Study tokenization

To generalize over HPO problems of different parameter sizes, types, and metadata, we propose to serialize the study as a one-dimensional textual sequence, also advocated in [26]. Unfortunately, a naive serialization approach, e.g. via JSON [43], will produce unnecessarily long sequences.

To improve scalability, we compress the textual representation of metadata $m$ by removing redundant phrases and punctuation (e.g., "parameter", quotes) and encoding keywords (e.g., "name", "algorithm") and enumerating types (e.g. "DOUBLE") into single tokens.

For the historical sequence $\boldsymbol{h}$, we convert every DOUBLE and INTEGER parameter along with every function value into a single token, by normalizing and discretizing them into integers, with an quantization level of $Q = 1000$; e.g.

$$\bar{x} = \text{int}[x_{\text{norm}} \cdot Q], \text{ where } x_{\text{norm}} = (x - x_{\min})/(x_{\max} - x_{\min}). \tag{1}$$

The range of $x$ is defined by the search space and the range of $y$ is obtained from observed values in $\boldsymbol{h}$. For other types, we use the index in their value set.

The shortened text string is then converted to a sequence of tokens via the SentencePiece tokenizer [44] (see Table 2 for an example). Every trial is represented by text, which is represented as a sequence of normalized and quantized tokens, $\left[\bar{x}_t^{(1)}, \ldots, \bar{x}_t^{(D)}, \star, \bar{y}_t, "|"\right]$, where the token $\star$ separates parameter and function values and "|" separates trials. See Appendix A.2 for further details on tokenization.

## 4.2 Model and training loss

After tokenization, the converted historical sequence is as follows:

$$\bar{\boldsymbol{h}}_t = \left[\bar{x}_1^{(1)}, \bar{x}_1^{(2)}, \ldots, \bar{x}_1^{(D)}, \star, \bar{y}_1, "|", \ldots, \bar{x}_t^{(1)}, \bar{x}_t^{(2)}, \ldots, \bar{x}_t^{(D)}, \star, \bar{y}_t\right]. \tag{2}$$

We can now apply a Transformer model to learn the conditional distribution of tokens in $\bar{h}$ using the chain rule, given the metadata $\bar{m}$, as depicted in Fig. 1. The joint distribution is presented in Appendix D.1.

Given a dataset $\mathcal{D}$ of hyperparameter optimization studies, we train the OPTFORMER by maximizing the weighted log-likelihood for each study $(m, \boldsymbol{h}) \sim \mathcal{D}$:

$$\mathcal{L}(\theta; m, \boldsymbol{h}) = \sum_n w_n \log P_\theta(\bar{h}^{(n)} | \bar{m}, \bar{\boldsymbol{h}}^{(1:n-1)}), \tag{3}$$

with $w_n = 0$ if $\bar{h}^{(n)} \in \{\star, "|"\}$ and $w_n = 1$ otherwise. That is, we mask out the separator tokens ($\star$, "|") and predict parameter $\bar{\boldsymbol{x}}$ and function tokens $\bar{y}$ only. Note that $\bar{h}^{(n)}$ denotes the $n$-th token, that is the $n$-th element of the list in Equation (2), and $\bar{\boldsymbol{h}}^{(1:n-1)}$ denotes all tokens up to the $(n-1)$-th token. Further details and data augmentations are provided in Appendix D.2.

## 4.3 Inference and decoding

**Parameter prediction:** To decode the predicted parameter token $\bar{x}_t^{(d)}$ back to its original parameter range, we truncate the output distribution to the vocabulary range corresponding to valid parameter values $[0, Q)$ and reverse our tokenization procedure in Section 4.1. For a `DOUBLE` or `INTEGER` parameter $x$, we use a piecewise constant distribution:

$$p_\theta(x|\ldots) = Q \cdot P_\theta(\bar{x}|\ldots)/(x_{\max} - x_{\min}), \text{ if } x \in [x_{\min}, x_{\max}], \text{ otherwise } 0. \tag{4}$$

For all other parameter types, $\bar{x}$ corresponds to the index of the set of feasible values. Putting these together, we may now sample parameter $\boldsymbol{x}_t$ from the model's prior distribution and thus define an HPO policy:

$$\pi_{\text{prior}}(\boldsymbol{x}_t|m, \boldsymbol{h}_{t-1}) = \prod_{d=1}^{D} p_\theta(x_t^{(d)}|m, \boldsymbol{h}_{t-1}, \boldsymbol{x}_t^{(1:d-1)}). \tag{5}$$

As we use a supervised learning loss, we expect $\pi_{\text{prior}}$ to approximate the behavior policy $\pi_b$.

Note that traditional BO algorithms require running Bayesian inference and then conducting a global search in the hyperparameter space with an acquisition function. Thus the runtime complexity of making one hyperparameter suggestion is cubic in $t$ for a typical GP-based BO method that performs ARD each iteration [45]. In contrast, generating one suggestion by the OPTFORMER consists of decoding $D$ parameter tokens with an input sequence of $(D+3)t$ tokens, which are then parsed into the $D$ parameter values, producing a runtime of $\mathcal{O}(D^2 t)$ linear in $t$, with proper caching.

**Function prediction:** To decode the real-valued function $y_t$ from the discrete distribution $P_\theta(\bar{y}_t|\bar{m}, \bar{\boldsymbol{h}}_{t-1}, \bar{\boldsymbol{x}}_t)$, we construct the same piecewise constant distribution as in Eq. (4) with the range $[y_{\min}, y_{\max}]$ used in tokenization. Note that the limited support of $y$ will not be a concern for HPO when either the range is known or we set the range large enough compared to observed values. For more general use as a few-shot function prediction model, one could consider adopting the Riemann Distribution in [40], which supports an unbounded range.

**Augmented HPO policies with function prediction:** At best, the learned policy $\pi_{\text{prior}}$ can only perform as well as the original policy $\pi_b$ when using behavioral cloning. However, we can take advantage of the model's simultaneous function prediction ability to improve the policy with model-based planning or offline RL techniques. While a comprehensive study of policy improvements for Transformers is out of the scope of this work, we consider here a simple yet effective policy improvement operator: sampling $M = 100$ candidate suggestions from $\pi_{\text{prior}}$ and choosing the suggestion with the highest score defined by an acquisition function $u(\cdot)$ as follows:

$$\pi_u(\boldsymbol{x}_t|m, \boldsymbol{h}_{t-1}) = \underset{\{\boldsymbol{x}^{(i)}\}_{i=1}^{M}}{\operatorname{argmax}} u(p_\theta(\cdot|m, \boldsymbol{h}_{t-1}, \boldsymbol{x}^{(i)})), \text{ with } \boldsymbol{x}^{(i)} \overset{\text{i.i.d.}}{\sim} \pi_{\text{prior}}(\boldsymbol{x}|m, \boldsymbol{h}_{t-1}). \tag{6}$$

Common acquisition functions include Expected Improvement (EI), Probability of Improvement (PI), Upper Confidence Bound (UCB), and Thompson Sampling, see for example [46]. At a high level, this approach to combining imitated policies with function prediction is reminiscent of the idea behind the offline RL approach of BCQ [47].

Because we apply a linear mapping from the original $y$ value to the quantized value $\bar{y}$ before discretization, we can simply define the acquisition functions on the discrete distribution $P_\theta(\bar{y}|\bar{m}, \bar{\boldsymbol{h}}_{t-1}, \bar{\boldsymbol{x}}_t)$ as follows:

$$u_{\text{EI}}(\boldsymbol{x}|\bar{y}^*) = \mathbb{E}_{P_\theta(\bar{y}|m, \boldsymbol{h}_{t-1}, \boldsymbol{x})} \left[\max(\bar{y} - \bar{y}^*, 0)\right], \tag{7}$$

$$u_{\text{UCB}}(\boldsymbol{x}|\alpha) = \text{Quantile}(P_\theta(\bar{y}|m, \boldsymbol{h}_{t-1}, \boldsymbol{x}_t), \alpha), \tag{8}$$

$$u_{\text{PI}}(\boldsymbol{x}|\bar{y}^*) = \sum_{\bar{y} > \bar{y}^*} P_\theta(\bar{y}|m, \boldsymbol{h}_{t-1}, \boldsymbol{x}), \tag{9}$$

$$u_{\text{TS}}(\boldsymbol{x}) = \bar{y}, \text{ with } \bar{y} \sim P_\theta(\bar{y}|m, \boldsymbol{h}_{t-1}, \boldsymbol{x}_t), \tag{10}$$

where $\bar{y}^* = \max_{\tau \leq t-1} \bar{y}_\tau$ in EI and PI is the threshold to measure improvement. We define the UCB acquisition function with a quantile parameter $\alpha$. Our TS acquisition is defined as a sampled function value at a given location from the marginal predictive distribution. It is inspired by the traditional Thompson Sampling method [45] but different in that the correlation between different locations is ignored.

# 5 Data

Training the OPTFORMER requires HPO studies with optimization trajectories. The most natural dataset we possess is the entire Google Vizier [2] database, one of the world's largest collections of real world hyperparameter tuning studies, which we denote as **RealWorldData**. There are around 750K studies, each with on average 300 trials, covering a vast class of production and machine learning applications at Google, ranging from vision, speech, NLP and robotics, and representing one of the most representative distributions of HPO tasks for machine learning models in practice. These studies were generated with a mixture of non-adaptive, evolutionary, and BO algorithms. However, as the dataset does not contain sufficient algorithm information, we have to treat the corresponding behavior policy as a randomly mixed algorithm $\pi_b$.

In addition, we create two new datasets based on public benchmarks. **HPO-B** is the largest public benchmark for HPO containing about 1.9K tuning tasks, most of which use one of 16 shared search spaces. In the continuous evaluation setting, it fits an XGBoost model to the trial data of every tuning task as the objective function. For further control over specific function dimensions and properties, we use the blackbox optimization benchmark **BBOB** [48], consisting of 24 types of synthetic functions with customizable properties (dimension sizes, rotations, shifts, discretizations, noise types) we randomize over.

For each of the two public benchmarks (HPO-B and BBOB), we apply a fixed set of 7 HPO algorithms to generate a dataset of optimization trajectories. In contrast to RealWorldData, we specify the algorithm name in the metadata $m$ as part of the conditioning input for our model. The controlled algorithms used are: (1) Grid Search, (2) Shuffled Grid Search, (3) Random Search, (4) Regularized Evolution [49], (5) Hill-Climbing, (6) Eagle Strategy [50], and (7) Vizier's GP-UCB [2]. Appendix B contains detailed explanations of the algorithms.

Table 3: Offline training datasets considered in this study. More details are given in Appendix C along with examples of studies in Table 5.

|  | ("R") **RealWorldData** | ("H") **HPO-B** | ("B") **BBOB** |
|---|---|---|---|
| #Studies | 750K | 10M | 10M |
| #Trials / study | 300 (on average) | 120 | 300 |
| Study source | Google's database | Generated | Generated |
| $\pi_b$ | Mixed | Controlled | Controlled |
| Obj. Functions | HPO tasks | HPO tasks | Synthetic |
| Search space | Different per task | 16 shared search spaces | Randomized |

# 6 Experiments

We train a single Transformer model with 250M parameters on the union of the three datasets described above, RealWorldData, HPO-B, and BBOB (hyperparameter details in Appendix D.2).

Each dataset contains a corresponding "test" set of functions, either using synthetic functions (BBOB) or fitting a machine learning model to obtain the objective (RealWorldData, HPO-B). We evaluate mainly on the two natural HPO benchmarks, RealWorldData and HPO-B. The train/test subsets of RealWorldData are split temporally to avoid information leak (see Appendix C for details).

To aggregate results across functions with different output scaling, we normalize all the test functions. This is standard practice in the literature [2, 5, 51–54]. We define our performance metric at trial $t$ as the best-so-far normalized function value $\max_{i \in \{1:t\}} (y_i - y_{\text{rand}})/(y_{\text{max}} - y_{\text{rand}})$, where $y_{\text{rand}}$ is the median of function values randomly sampled in the search space to be robust to outliers, and $y_{\text{max}}$ is the maximum, if known, or best value found by any algorithm. For the HPO-B benchmark, we use the recommended bounds provided in [5]. We also consider other metrics when comparing different algorithms in Appendix E.3, including the performance profile and average ranking. We find our results are consistent over different metrics.

Because the OPTFORMER is trained to predict the conditional distributions of parameter and function values, we would like to answer the following questions when evaluating on unseen test problems:

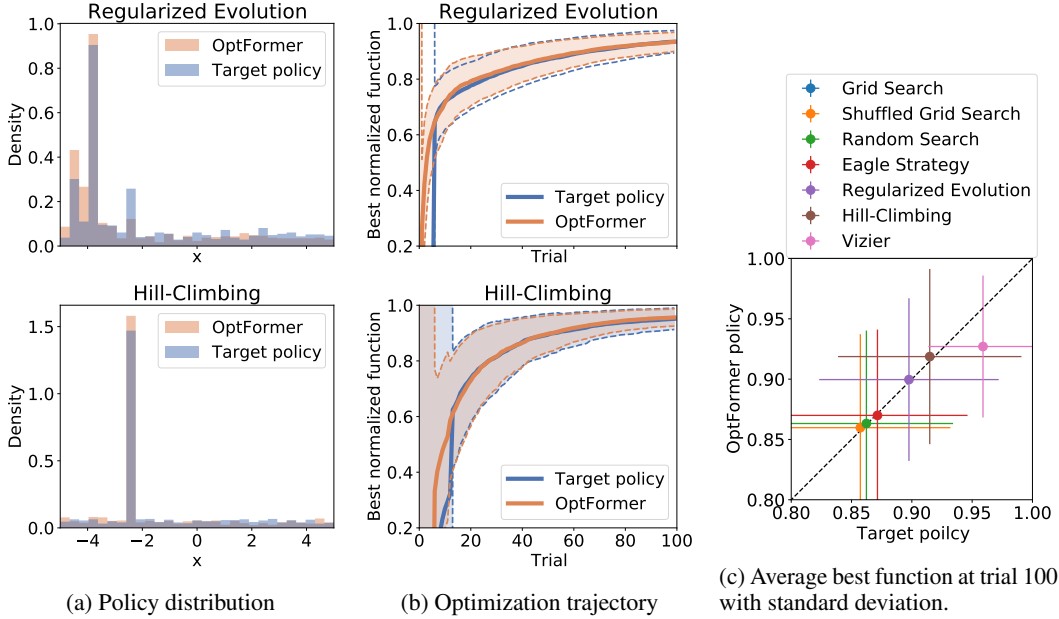

Figure 2: Comparing the performance of different algorithms outputted by the OPTFORMER conditioned on the corresponding algorithm's name.

1. Can the OPTFORMER learn to imitate multiple HPO algorithms with one model? (Section 6.1)

2. Can the OPTFORMER learn a good prior over hyperparameter response functions? (Section 6.2)

3. Is the OPTFORMER a competitive approach for HPO? (Section 6.3)

### 6.1 Imitating HPO policies

We first evaluate how well the OPTFORMER can learn the conditional distribution of parameter suggestions given by the behavior policies in the dataset, and how well it can imitate multiple algorithms. As the algorithm's name is contained in the metadata $m$, we can modify the behaviour of the policy $\pi_{\text{prior}}(\boldsymbol{x}_{t+1}|m, \boldsymbol{h}_t)$ simply by altering this variable. Fig. 2a compares two different policies to the OPTFORMER, when it is conditioned on the corresponding policy name. We observe a good match between the imitated algorithms and the OPTFORMER (additional algorithms are shown in Appendix E.1).

In Fig. 2b we run target policies on the BBOB dataset's test functions and compare the optimization trajectories of the algorithms and the OPTFORMER. In Fig. 2c we compare the average and standard deviation of the best normalized function values at trial 100. Our model imitates most algorithms very accurately in both the mean and variance except for the most complicated algorithm, Vizier, where $\pi_{\text{prior}}$ is slightly worse in the LUNACEK benchmark. We expand on this in Appendix E.1. Because Vizier is the best performing HPO algorithm among all considered, the OPTFORMER will imitate Vizier faithfully, although not perfectly, in the following experiments.

### 6.2 Learning priors for hyperparameter response functions

In this section, we assess the OPTFORMER's ability to learn the conditional distribution of the function value as a few-shot function regressor. Specifically, for every function in each test dataset, we repeatedly sample up to 200 random trials $(\boldsymbol{x}_1, y_1, \ldots \boldsymbol{x}_t, y_t), t \leq 200$, and predict the conditional distribution $p(y_t|\boldsymbol{x}_1, y_1, \ldots, \boldsymbol{x}_t)$. We compare with a GP model with output warping — details provided in Appendix B. We report the log-predictive likelihood $\log p(y_t|\boldsymbol{x}_t, \ldots)$ in Table 4.

As uncertainty estimation is important for HPO, we also evaluate how well the function predictive distribution is calibrated. When a predictive distribution $p_\theta(y|\ldots)$ matches the true distribution, the estimated CDF $F(y) = \int_{-\infty}^{y} p_\theta(y'|\ldots)dy'$ will be uniformly distributed. In Fig. 3, we plot the cumulative histogram of $F(y)$ on RealWorldData test set and check the deviation from the diagonal line to assess goodness-of-fit as proposed by Rosenblatt [55]. The OPTFORMER has a smaller

Table 4: Log-predictive likelihood (with 1-std. standard error, higher is better (↑)) and ECE (percentage of error, lower is better (↓)) on RealWorldData and HPO-B test sets.

| Model | Log-predictive likelihood ↑ | |
| --- | --- | --- |
| | RealWorldData | HPO-B |
| GP | 0.83(0.06) | 4.03(0.04) |
| OPTFORMER | **2.12 (0.05)** | **6.16 (0.04)** |
| Model | ECE (percent %) ↓ | |
| | RealWorldData | HPO-B |
| GP | 5.34 (0.06) | 2.39 (0.05) |
| OPTFORMER | **1.11 (0.02)** | **1.89 (0.01)** |

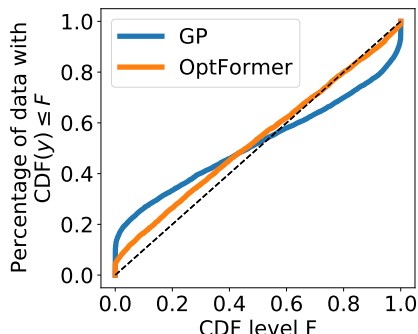

Figure 3: Cumulative histogram of predicted $CDF(y)$ on RealWorldData test set.

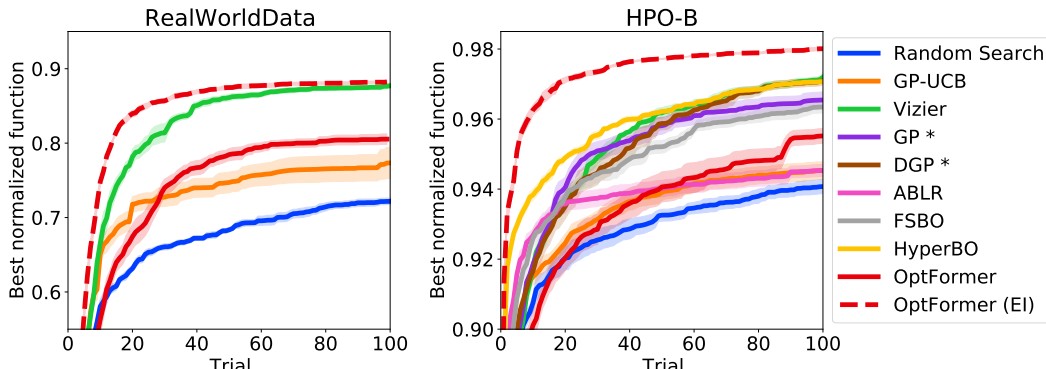

Figure 4: Higher is better. Best normalized function value averaged over 16 RealWorldData test functions (left) and over 86 HPO-B test functions (right) with 1-std confidence interval from 5 runs. GP* and DGP* results are provided by [5]. The transfer learning methods ABLR, FSBO and HyperBO cannot be applied to RealWorldData.

deviation than the GP almost across the entire range. We also compare calibration performance using the expected calibration error (ECE) [27]. Readers are referred to [27] and Appendix E.2 for a detailed explanation of ECE. We observe from Table 4 that the OPTFORMER achieves better predictive likelihood and ECE than the GP on both datasets.

## 6.3  Augmenting a prior policy with function prediction

We evaluate the OPTFORMER as a hyperparameter optimization algorithm on two benchmarks, RealWorldData and HPO-B. We compare our prior policy, the OPTFORMER, and an augmented policy with Expected Improvement, the OPTFORMER (EI), against standard HPO baselines, including Random Search, our implementation of GP-UCB, and the well-tuned Vizier service. For HPO-B, we also include the GP (not to be confused with our GP-UCB) and DGP (GP with deep kernel) baseline results provided by the original paper [5]. Additionally, we include three recent transfer-learning methods based on multi-task GP models: ABLR [12, 56], FSBO [7], and HyperBO [57, 58] (implementation details in Appendix B). Please note that all of these transfer learning methods require learning GPs on multiple tasks sharing the same search space. Therefore, none of them apply to the RealWorldData benchmark where every study has its own search space.

We show the trajectory of the best normalized function value averaged over all functions from each benchmark in Fig. 4. While the prior policy returned by the OPTFORMER does not perform as well as Vizier, it is comparable or slightly better than our GP-UCB baseline and ABLR.

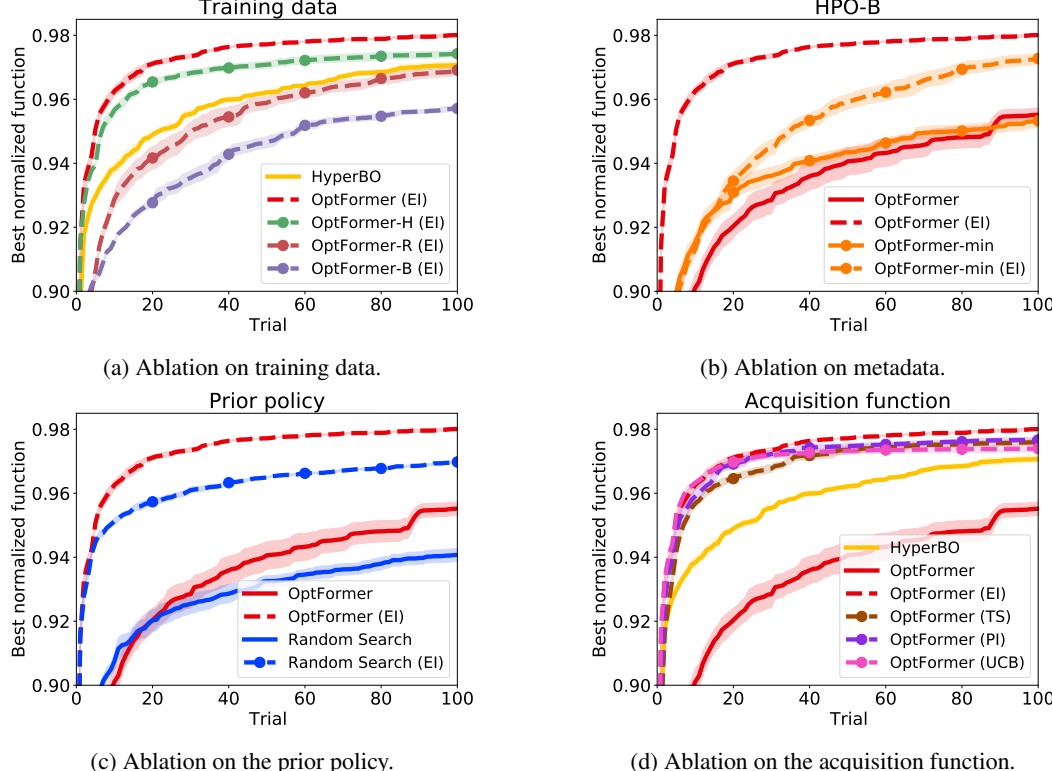

| (a) Ablation on training data. | (b) Ablation on metadata. |

| (c) Ablation on the prior policy. | (d) Ablation on the acquisition function. |

Figure 5: Best normalized function values averaged over HPO-B test functions with 1-std confidence interval. Ablation curves are shown with ◯ markers. (a) The more similar the training dataset, the better the transfer. Here, the suffix with "H", "R", "B" indicates training on HPO-B, RealWorldData, and BBOB respectively. (b) Removing the majority of metadata hurts function prediction. (c) The prior policy improves performance with or without the Expected Improvement acquisition function. (d) All acquisition functions provide a significant improvement.

The most significant improvement is achieved when we augment our prior policy with the Expected Improvement acquisition function. The resulting OPTFORMER (EI) outperforms all baselines across the board on both benchmarks. This illustrates that the OPTFORMER is able to learn the distribution of functions in the meta-training split and transfers to the meta-testing split.

It is worth noting that to run 100 trials for about half of the test functions, the required history token sequence is longer than the 1024-token length used in training, with the maximum length about twice the training horizon. The superior performance of the OPTFORMER (EI) thus demonstrates its good generalization performance beyond the optimization horizon it is trained for.

## 6.4 Ablations

We provide further ablations on three important components for our policy:

**Training dataset.** To understand the impact of the training datasets on the OPTFORMER, we train three variants on individual datasets (OPTFORMER-"R","H","B" respectively for RealWorldData, HPO-B, BBOB) and study their transfer learning performances on HPO-B. Fig. 5a verifies that training with in-domain data ("H") gives better performance than training over the more diverse across-domain RealWorldData HPO dataset ("R"), which is better than training over the synthetic BBOB data ("B"). Nonetheless, training on RealWorldData is enough to give comparable performance to the best transfer learning baseline at the end of 100 trials. Lastly, training on all of the datasets (OPTFORMER) gives a further advantage over OPTFORMER-H. This suggests that more data does not hurt the model's performance but rather may improve it, even if the extra data is out-of-domain.

**Meta-data** $m$**.** We have demonstrated how the OPTFORMER's behavior can be controlled by the algorithm name in metadata $m$ in Section 6.1. Here we study whether the OPTFORMER learns to depend on other meta information. At inference time, we provide minimum information in $m$ (OPTFORMER-min) by excluding all textual information and parameter value ranges. We only keep necessary information such as parameter types and algorithm names. Fig. 5b shows that the prior policy of OPTFORMER-min performs comparably with the OPTFORMER, partly due to the use of data augmentation (see Appendix D.2). The augmented policy OPTFORMER-min (EI) (dashed orange) improves upon the prior policy but is significantly *worse* than the full model, suggesting that the missing metadata impacts the model's predictions on function values.

**Prior policy.** Section 6.3 demonstrated the benefit of adding an acquisition function to the prior policy. A natural question is whether a good prior policy is needed at all. In Fig. 5c, we replace the prior policy in the OPTFORMER (EI) with random search (Random Search (EI), dashed blue line). While adding Expected Improvement still improves this random search policy's performance, the best method requires both a good prior policy and the acquisition function.

**Choice of acquisition function.** In Fig. 5d, we compare the Expected Improvement (EI) with Thompson Sampling (TS), Probability of Improvement (PI), and Upper Confidence Bound (UCB) with a confidence level of 0.9. We observe that the prior policy is improved by all the acquisition functions. Particularly, OPTFORMER (EI) is the best among all the choices though the difference is relatively small compared to the advantage over other baselines and OPTFORMER prior policy. We provide additional analysis with results on both the RealWorldData and HPO-B datasets, as well as other evaluation metrics in Appendix E.4.

# 7 Limitations and future extensions

We list a few limitations of this work and discuss some potential extensions. (1) We did not consider parameters that do not always apply or are subject to dynamic constraints depending on other parameter values. Such parameters are common in AutoML [59] and NAS applications [60]. Our work can be extended to support these applications, by providing the conditional specifications as text in metadata $m$. (2) We also considered only sequential optimization with a batch size of 1. To support parallel suggestions, one could apply random masking to input function value observations to simulate scenarios with parallel pending trials [33]. (3) While we trained the Transformer to clone the behavior policy offline, there are extensive literature on offline RL [29] that could be applied here [25, 47, 61–64]. One could also consider meta-training acquisition functions as in [31] within the same model and online fine-tuning as in [7, 41]. (4) We considered a single objective function, though multiple objectives can be easily included by outputting multiple function tokens in a trial. (5) The maximum sequence length is limited by the quadratic memory size requirement of a Transformer, which could be mitigated with more scalable architecture variants such as Performer [65].

# 8 Conclusion

We presented first step to learning a universal Transformer model for hyperparameter optimization from large scale datasets containing tuning experiments with vastly different search spaces and experiment descriptions. By training on a diverse set of synthetic and real-world tuning trajectories, we demonstrated the capacity of a single Transformer model to imitate 7 fundamentally different HPO policies, learn to make well calibrated few-shot function predictions, and provide competitive optimization performance on unseen test functions comparable with the existing, long-tried GP-based baselines. Many extensions are readily conceivable for future exploration.

## Acknowledgments

We would like to thank Chris Dyer, Luke Metz, Kevin Murphy, Yannis Assael, and Esteban Real for providing valuable feedback during their reviews of this paper. We further thank Sebastian Pineda Arango for technical discussions on the HPO-B benchmark and Christof Angermueller on biological benchmarks. In addition, we thank Daniel Golovin, Daiyi Peng, Yingjie Miao, Jack Parker-Holder, Jie Tan, Lucio Dery, and Aleksandra Faust for multiple useful conversations.

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
