# APPENDIX

## A   Preprocessing and tokenization details

### A.1   Search space primitives

Below are the exact descriptions of the hyperparameter primitives used to define a given $\mathcal{X}^{(d)}$.

- `Double:` Specifies a continuous range of possible values in the closed interval $[x_{\min}, x_{\max}]$ for some real values $x_{\min} \leq x_{\max}$.
- `Integer:` Specifies an integer range of possible values in $[x_{\min}, x_{\max}] \in \mathbb{Z}$ for some integers $x_{\min} \leq x_{\max}$.
- `Discrete:` Specifies a finite, ordered set of values from $\mathbb{R}$.
- `Categorical:` Specifies an unordered list of strings.

### A.2   Data preprocessing and tokenization

We list out the full set of preprocessing steps (from Section 4.1) below:

- Omit parameter and metric names in all trials, remove redundant keywords (`"parameter"`, `"trial"`, etc.), **order trial parameters** according to those in metadata $m$, and add keywords (e.g., "name", "algorithm") and enumerating types (e.g. "DOUBLE") in the tokenizer vocabulary so that the original keywords are encoded into single tokens.
    - List of keywords: name, metric, goal, type, algorithm, min_value, max_value, scale_type, categories.
    - Enumerating values for the parameter type: DOUBLE, INTEGER, DISCRETE, CATEGORICAL.
    - Enumerating values for the scale_type: LINEAR, LOG.
- Insert short separator symbols, e.g. $\star$ between parameter/metrics in a trial, "|" between trials, and "&" between experiment description and parameter configurations in metadata.
- Convert all values in history $h$ to single integers.
    - Represent discrete and categorical parameters with their index in the set of values.
    - Normalize float and integer parameter values in $x_t^{(d)}$ with their value range and the function values $y_t$ with their minimum and maximum seen values in the entire study. Then quantize the normalized values to an integer, e.g., "0.12345" $\rightarrow$ "123" with a quantization level of $Q = 1000$. More formally, we apply the following transformation $q(\cdot)$:

$$q(z) = \text{int}[z_{\text{norm}} * Q], \text{ where } z_{\text{norm}} = (z - z_{\min})/(z_{\max} - z_{\min}) \qquad (11)$$

The shortened text string is then converted to a sequence of tokens via the SentencePiece tokenizer [44] with a vocabulary of 33000 words. Quantized numbers in $h$ are always converted into single tokens. As long as $Q$ is sufficiently large, there is no concern from the loss of precision over numerical quantizations, and thus the serialized study contains nearly the same amount of information as the original data. For comparison, the naive tokenization for the example of Table 1 with $t = 100$ trials will produce 8221 tokens which can overload GPU memory, while our proposed tokenization will only produce 584 tokens, a 14x reduction.

# B Algorithm and baseline details

## B.1 Dataset algorithms

**Grid Search:** `DOUBLE` parameters are first log-transformed if specified. They are then converted into `DISCRETE` parameters by discretizing their ranges into 100 equidistant points. Suggestions are outputted using lexicographic ordering from the cartesian product over all parameters' feasible points. The traversal order follows the alphabetical ordering of parameter names. That is, given two parameters `"Foo"` and `"Bar"`, both in [0,1] range, the sequence of trials looks like: `{"Foo": 0, "Bar":0}` , `{"Foo": 0, "Bar":0.01}`, ..., `{"Foo": 0, "Bar":1}`, `{"Foo": 0.01, "Bar":0}`, `{"Foo": 0.01, "Bar":0.01}`, ....

**Shuffled Grid Search:** Shuffled grid search is the same as Grid Search in how it handles `DOUBLE` parameters. Instead of traversing the grid in a deterministic order, it selects without replacement a random point from the grid at each iteration.

**Regularized Evolution [49]:** In summary, this algorithm at every iteration randomly selects a tournament subset from the current population, and mutates the argmax member of the tournament. When inserting a new trial, the oldest trial will be removed. We use a population size of 25 and tournament size of 5. The mutation operation uniformly at random selects one of the parameters $x^{(r)}$ from $\boldsymbol{x}$, and mutates $x^{(r)}$ based on the following: for `DOUBLE`, `INTEGER`, the new value is uniformly sampled from $\left[x_{\min}^{(r)}, x_{\max}^{(r)}\right]$, while for `DISCRETE`, `CATEGORICAL`, the new value is uniformly sampled from the feasible list.

**Hill Climbing:** This is a naive implementation, where at every iteration $t$, the current $\boldsymbol{x}_{\text{pivot}}$ is mutated (using the same operation as Regularized Evolution) to $\boldsymbol{x}_{\text{mutated}}$, and evaluated. If $f(\boldsymbol{x}_{\text{mutated}}) > f(\boldsymbol{x}_{\text{pivot}})$, then we reassign $\boldsymbol{x}_{\text{pivot}}$ to be the mutated $\boldsymbol{x}_{\text{mutated}}$. An extension of this method can be "batched", as seen in [66], although we not include this for the sake of clarity and presentation.

**Eagle Strategy [50]:** Eagle strategy is a metaheuristics algorithm that is a slight variation of Particle Swarm Optimization [67].

The algorithm is originally formulated for continuous search spaces only. The reason is that it involves a subroutine (`move` step) where we take a convex combination of a particle (called *firefly* in [50]) and another particle that has a better objective value. Mathematically, given two particle vectors $\boldsymbol{x}$ and $\boldsymbol{x}'$ and the coefficient $c \in [0, 1]$, the `move` step generates $c\boldsymbol{x} + (1 - c)\boldsymbol{x}'$.

The algorithm is extended to support `DISCRETE` and `CATEGORICAL` parameters by applying a separate `move` operation for each non-continuous dimension $d$:

$$\texttt{move}(x^{(d)}, x'^{(d)}, c, \alpha) = \begin{cases} x^{(d)} & \text{with probability } (1 - \alpha)c \\ x'^{(d)} & \text{with probability } (1 - \alpha)(1 - c) \\ \text{random value} & \text{with probability } \alpha \end{cases}$$

where $\alpha$ is a small perturbation coefficient that decreases in the dimension of the search space.

**Vizier [2]:** Vizier's default algorithm is available via Google Cloud as Vertex Vizier. We have contacted the authors of the algorithm and received the the following details on its implementation.

In summary, the algorithm uses a particular implementation of GP-UCB with trust regions. The GP regressor model consists of the following:

- $\alpha \sim \text{TruncatedLogNormal}$ controls the amplitude of Matern5/2 kernel.
- $\lambda_i \sim \text{TruncatedLogNormal}$ (i.i.d. for each dimension $i$) controls the length scale for the $i$-th dimension.
- $\sigma \sim \text{TruncatedLogNormal}$ controls the Gaussian noise.
- $z \sim \text{Normal}(0, \sigma)$ is the observation noise.
- $f \sim \text{GP}(\lambda, \alpha)$ is the function.

- $y(x) \sim f(x) + z$ is the noisy function.

where the prior distribution parameters are chosen depending on the user's estimate of the observation noise.

The algorithm then uses gradient descent with line search for step sizes to obtain the MAP estimate of $\alpha, \lambda$ and $\sigma$. Furthermore, the algorithm uses a variation of Eagle Strategy (explained above) to optimize the UCB acquisition function with coefficient of 1.8. In order to prevent overexploration that may result from the large UCB coefficient, the algorithm optimizes acquisition functions inside trust region. The trust region is the union of $L_\infty$-norm balls around explored points. The radius of the $L_\infty$-norm ball grows in the number of explored points. The algorithm also starts at the center of the search space (unless user specifies an alternative initial batch).

**GP-UCB:** It is the same as Vizier's GP-UCB, except for the model definition. We used the model definition from the github repository of the authors of "Heteroscedastic and Evolutionary Bayesian Optimisation solver" (HEBO) [68], the winner of 2020 Blackbox Optimization challenge [51]. It is worth noting that HEBO uses multi-dimensional acquisition functions derived from the GP model. The priors over hyperparameters are thus not tuned to optimize the performance of GP-UCB algorithm, which explains its suboptimal performance.

## B.2 Gaussian Process for uncertainty estimation

We use the same GP model as GP-UCB.

When comparing the function prediction performance with the OPTFORMER, we choose $[y_{\min}, y_{\max}]$ to normalize function value token based on the range of observed value in the sampled sequence $(\boldsymbol{x}_1, y_1, \dots \boldsymbol{x}_t, y_t)$, and therefore the real value of $y_t$ always resides in the prediction support of the OPTFORMER.

To compensate for the fact that GP's distribution is wider than the real support used by the Transformer, we truncate the GP's prediction into $[y_{\min}, y_{\max}]$ for a fairer comparison.

## B.3 Transfer learning baselines

We use the following methods as transfer-learning baselines for the HPO-B dataset from Section 6.3:

**ABLR [12, 56]:** BO with multi-task adaptive Bayesian linear regression. Our implementation of ABLR is equivalent to a GP with 0 mean and a dot-product kernel with learned basis functions. We use a neural net (NN) with $(128, 128)$ hidden layers and tanh activation as the basis functions. We then train ABLR by optimizing the negative log likelihood (NLL) over NN weights $\theta$ as well covariance matrix $SS^\top$ and bias parameters $\delta^2$ that define the dot-product kernel $k$, i.e.

$$k(x, x') = \phi_\vartheta(\boldsymbol{x})^\top SS^\top \phi_\vartheta(\boldsymbol{x}') + \delta^2, \tag{12}$$

where matrix $S \in \mathbb{R}^{128 \times 256}$, basis function $\phi_\theta$ is parameterized by NN weights $\vartheta$ and $\delta \in \mathbb{R}$.

**FSBO [7]:** Bare-bone few-shot BO. We did not include data-driven initialization due to lack of reproducing details. Following [7], our implementation of FSBO is equivalent to BO using a GP with 0 mean and a squared-exponential kernel on top of a NN with $(128, 128)$ hidden layers and tanh activation functions. We train the NN weights as well as the parameters in the squared-exponential kernel.

**HyperBO [57, 58]:** BO with pre-trained GPs. Following [58], we pre-train a GP with Matérn32 kernel on top of a NN with one hidden layer of width $2 \times D$ and tanh activation functions. Here $D$ is the input dimension of the search space.

For training, we use the Adam optimizer with learning rate 0.001 and batch size 50 for all the transfer-learning baselines. Notice that these transfer-learning methods require "pre-training" a GP on the same search space. We sample 10000 random data points on each HPO-B surrogate functions from each search space. We train a separate GP for each search space.

# C Data details

## C.1 Dataset details

**RealWorldData dataset:** The RealWorldData dataset contains a total of 750K studies collected from Google Vizier users over a span of 5 years (starting from 2017), and each study has a variable number of trials. Since some user studies can potentially have an excessive number of trials (e.g. 10K+), for all studies we only consider the first 300 trials for experiments. Since the dataset also includes Google employee usernames, we made sure to anonymize every study first.

We split the dataset in temporal order to avoid information leak, use most studies for training, and select 16 studies generated by a different set of users for testing. All training studies were generated before Feb 2020. The test studies were created by users who started to use the hyperparameter tuning service after that date. To bootstrap these studies into actual functions to be evaluated, we fit a GP for each study and output the function value as the GP's median function (due to the use of output warping).

**HPO-B dataset:** For HPO-B dataset, a tuning task is identified with a (search space id, dataset id) pair, which refers to tuning the hyperparameters defined in a search space for some machine learning model trained on a dataset. we use the `"v3-augmented"` meta-training/validation/test splits that includes all the 16 test search spaces as well as less frequent search spaces in the meta-training split. There are uniquely 1730, 91, and 86 tasks for training, validation and testing respectively. For every tuning task, [5] fits an XGBoost model to the trial data of every tuning task as the objective function.

Similar to the BBOB dataset, we generate 10M, 500K studies for training and validation respectively, along with the same set of controlled algorithms. For each of the test tuning task, we run 5 optimizations each with a different initial set of observations provided in [5].

The HPO-B uses the Apache 2.0 open-source license.

**BBOB dataset:** The BBOB dataset contains a total of 10M studies for training, each containing exactly 300 trials. An additional 500K studies (using different randomization seeds) are used for validation. While the number of studies can be freely generated and effectively unlimited, we found that 10M studies were sufficient for the Transformer to train properly.

The functions we use for data are from [48], and consist of separable functions (`Sphere`, `Ellipsoid Separable`, `Rastrigin Separable`, `Bueche Rastrigin`, `{Linear Slope}`), moderately conditioned, potentially multi-modal functions (`Attractive Sector`, `Step Elllipsoid`, `{Rosenbrock Rotated}`), ill-conditioned functions (`Discus`, `Bent Cigar`, `Sharp Ridge`, `{Sum of Powers}`), multi-modal functions (`Weierstrass`, `Schaffers F7`, `Schaffers F7 Illconditioned`, `{Greiwank Rosenbrock}`), and functions with weak global structures (`Schwefel`, `Gallagher 21`, `Gallagher 101`, `Katsuura`, `{Lunacek}`). The functions noted with the extra "{}" are for testing and excluded from the training data. We apply significant randomization over the functions for both the training dataset and test-time evaluation. In order, we randomize the following:

- Function dimension $D$, which is uniformly selected from a range. For training data generation, this range is $[1, 20]$.

- Orthonormal rotation matrix $\Gamma$, which is applied to the input first, i.e. producing a new function $f'(\boldsymbol{x}) = f(\Gamma \boldsymbol{x})$.

- Shift vector $\boldsymbol{x}_{shift}$ which is also applied to the input first, i.e. producing a new function $f'(x) = f(x - \boldsymbol{x}_{shift})$, where $\boldsymbol{x}_{shift}$ has all of its coordinate-wise entries sampled from $[-4, 4]$, while the domain is $[-5, 5]$.

- Discretization, in which the parameter space $\mathcal{X}^{(d)}$ is uniformly at random chosen to be either a `DOUBLE`, `DISCRETE`, `CATEGORICAL` parameter. The `DOUBLE` parameter "discretization" is actually a no-op, as it allows the original continuous space $\mathcal{X}^{(d)} \subset \mathbb{R}$. Otherwise, a number $L$ of feasible points is uniformly selected from the range $[2, 8]$, and used to divide the original $[-5, 5]$ range into $L$ equally-spaced points. If `DISCRETE` was chosen, then the ordering of the grid points is preserved, otherwise if `CATEGORICAL` was chosen, then all of the gridpoints become effectively unordered strings.

- Noise Type, in which one of 10 noise settings (including no noise) is uniformly chosen. Noise consists of either Gaussian (multiplier sampled from a random Gaussian of varying scale is applied), Uniform (multiplier sampled from uniform distribution of varying scale is applied), or Cauchy (additive noise which only occurs at a probabilistic frequency, with a varying fixed strength is applied).

For evaluation, we randomly sample 100 configurations for each of the five test functions, resulting in 500 optimization trajectories in total.

For BBOB, as all parameters are named as "x_i" with $i \in [0, D)$ and always have value range in $[-5, 5]$, significantly different from the other two datasets, we omit their parameter names and value in the metadata $m$ and only keep parameter type information.

Table 5: Example of studies in RealWorldData (left), BBOB (middle) and HPO-B (right).

```
"name": "gan1d 500 iters -          "name": "SCHAFFERS_F7",             "name": "5859_145853",
"2022-05-18"                        "algorithm": "gp",                  "algorithm": "GP UCB",
"parameter": {                      "parameter": {                      "parameter": {
  "name": "learning_rate",            "name": "x0",                       "name": "minsplit",
  "min_value": 1e-06,                 "type": "CATEGORICAL",              "max_value": 60.0,
  "max_value": 0.01,                  "categories": ["0.0", "5.0",        "type": "DOUBLE",
  "type": "DOUBLE",                                  "-5.0"],             "scale_type": "LINEAR",
  "scale_type": "LOG",             },                                  }
}                                   "parameter": {                      "parameter": {,
"parameter": {                        "name": "x1",                       "name": "minsplit.na",
  "name": "modifier",                 "min_value": -5.0,                  "max_value": 1.0,
  "min_value": 0.1,                   "max_value": 5.0,                   "type": "DOUBLE",
  "max_value": 1000000.0,             "type": DOUBLE,                   }
  "type": "DOUBLE",                   "scale_type": UNIT_LINEAR_SCALE,    "parameter": {
  "scale_type": "LOG",                                                    "name": "minbucket",
}                                   },                                    "min_value": 1.0,
"parameter": {                      "parameter": {                        "max_value": 60.0,
  "name": "weight_init_std",          "name": "x2",                       "type": "DOUBLE",
  "min_value": 0.01,                  "min_value": -5.0,                  "scale_type": "LINEAR",
  "max_value": 2.0,                   "max_value": 5.0,                 }
  "type": "DOUBLE",                   "type": DOUBLE,                   "parameter": {
}                                     "scale_type": UNIT_LINEAR_SCALE,    "name": "cp",
"parameter": {                                                           "min_value": 0.000100788830221,
  "name": "optimizer",             },                                    "max_value": 1.000092678873241,
  "type": "CATEGORICAL",           "parameter": {                        "type": "DOUBLE",
  "categories": "sgd",               "name": "x3",                       "scale_type": "LOG",
  "categories": "adam",              "type": DISCRETE,                 }
  "categories": "rmsprop",           "values": [-5.0, 5.0],             "parameter": {
}                                   },                                    "name": "maxdepth",
"goal": "MINIMIZE",                 "parameter": {                        "max_value": 29.0,
"max_num_trials": 500,                "name": "x4",                       "type": "DOUBLE",
"metric": "",                         "type": CATEGORICAL,                "scale_type": "LINEAR",
"observation_noise": "HIGH",          "categories": ["5.0",             }
"trial": {                               "-1.66666666667",              "parameter": {
 "parameter": {                                       "-5.0",            "name": "maxdepth.na",
   "learning_rate": 0.0001,            "1.666666666667"],               "max_value": 1.0,
   "modifier":                      },                                    "type": "DOUBLE",
         316.2277660168381,         "parameter": {                      }
   "optimizer": "sgd",                "name": "x5",                    "observation_noise": AUTOMATIC,
   "weight_init_std": 1.005,          "min_value": -5.0,               "metric": "objective_value",
 }                                    "max_value": 5.0,                "goal": "MAXIMIZE"
 "metric": {                          "type": DOUBLE,                  "trial": {
   "": -0.946908021738347,            "scale_type": UNIT_LINEAR_SCALE,  "parameter": {
 }                                                                        "minsplit": 4.0,
}                                   }                                     "minsplit.na": 0.0,
"trial": {                          "metric": "",                         "minbucket": 18.0,
 "parameter": {                      "goal": MAXIMIZE,                    "cp": 0.7342895964927976,
   "learning_rate": 0.000504,        "observation_noise": HIGH           "maxdepth": 3.0,
   "modifier":                       "trial": {                           "maxdepth.na": 0.0,
         12.346786652749216,         "parameter": {                     }
   "optimizer": "rmsprop",            "x0": "0.0",                      "metric": {
   "weight_init_std":                 "x1": 0.0,                         "objective_value": 0.500024080276,
         1.2192566347109868,          "x2": 0.0,                        }
 }                                    "x3": 5.0,                        }
 "metric": {                          "x4": "-5.0",                     "trial": {
   "": -1.5144472008077585,           "x5": 0.0,                         "parameter": {
 }                                   }                                     "minsplit": 8.0,
}                                    "metric": {                           "minsplit.na": 0.0,
...                                   "": -334.4782223514127,              "minbucket": 32.0,
                                     }                                     "cp": 0.30972302652187583,
                                    }                                     "maxdepth": 4.0,
                                    "trial": {                             "maxdepth.na": 0.0,
                                     "parameter": {                      }
                                      "x0": "5.0",                      "metric": {
                                      "x1": -1.9867479768748013,          "objective_value": 0.50002408028,
                                      "x2": -1.7665621302793095,         }
                                      "x3": -5.0,                       }
                                      "x4": "1.666666666666667",        ...
                                      "x5": -1.7634306558106605,
                                     }
                                     "metric": {
                                       "": -323.84900527589326,
                                     }
                                    }
                                    ...
```

# D Model and training details

The open-sourced T5 model codebase we use can be found at https://github.com/google-research/t5x.

## D.1 Conditional probability decomposition

From Section 4.2, the joint distribution of the optimization history $h$ conditioned on metadata $m$ can be written using the chain rule as

$$
P(\bar{\boldsymbol{h}}|\bar{m}) = P\left(\bar{x}_1^{(1)}, \bar{x}_1^{(2)}, \ldots, \bar{x}_1^{(D)}, \star, \bar{y}_1, "|", \ldots, \bar{x}_T^{(1)}, \bar{x}_T^{(2)}, \ldots, \bar{x}_T^{(D)}, \star, \bar{y}_T|\bar{m}\right)
$$
$$
= \prod_{t=1}^{T}\left(\prod_{d=1}^{D} P\left(\bar{x}_t^{(d)}|\bar{m}, \bar{\boldsymbol{h}}_{t-1}, \bar{\boldsymbol{x}}_t^{(1:d-1)}\right)\right) P\left(\star|\bar{m}, \bar{\boldsymbol{h}}_{t-1}, \bar{\boldsymbol{x}}_t\right) P\left(\bar{y}_t|\bar{m}, \bar{\boldsymbol{h}}_{t-1}, \bar{\boldsymbol{x}}_t\right) P\left("|"|\bar{m}, \bar{\boldsymbol{h}}_t\right)
$$
$$(13)$$

We note that this correctly formalizes the prediction of objects we are most interested in, which are parameter values $P\left(\bar{x}_t^{(d)}|\bar{m}, \bar{\boldsymbol{h}}_{t-1}, \bar{\boldsymbol{x}}_t^{(1:d-1)}\right)$ and function values $P\left(\bar{y}_t|\bar{m}, \bar{\boldsymbol{h}}_{t-1}, \bar{\boldsymbol{x}}_t\right)$.

## D.2 Training

During training, the encoder (denoted as $\mathbf{E}_\theta$) input sequence length is selected to be the maximum length of the tokenized metadata $\bar{m}$ from a dataset, ranging from 256 to 1024. The decoder (denoted as $\mathbf{D}_\theta$) input sequence is fixed at 1024, which means it can model up to $1024//(D+3)$ trials where $D$ is the number of parameters per trial. We use Adam optimizer with a rsqrt learning rate schedule and a mini-batch size of 256, and train each model up to 1M steps, with early stopping according to the validation loss. Each model is trained with a 4x4 TPU-v3 slice.

Thus the prediction for $\bar{h}^{(n)}$ is:

$$
P_\theta\left(\bar{h}^{(n)}\Big| m, \bar{\boldsymbol{h}}^{(1:n-1)}\right) = \text{SoftMax}\left[\mathbf{D}_\theta(\mathbf{E}_\theta(\bar{m}), \bar{\boldsymbol{h}}^{(1:n-1)})\right] \tag{14}
$$

## D.3 Data augmentation

We adopt the following three data augmentations to reduce overfitting to the offline datasets:

1. In order for the model to be invariant to parameter ordering, we apply random parameter permutations over metadata $\bar{m}$ and every suggestion $\bar{\boldsymbol{x}}_t$.
2. In order for the model to be robust to a different normalization range given a new function, we apply random scaling and shifting to the normalized function value $y_{\text{norm}} = (y - y_{\text{min}})/(y_{\text{max}} - y_{\text{min}})$ before quantization:

$$
y'_{\text{norm}} = y_{\text{norm}} * s + c, \ s \sim \text{Uniform}[0.3, 1], \ c \sim \text{Uniform}[0, 1-s] \tag{15}
$$

   and thus $y'_{\text{norm}} \in [c, c+s] \subseteq [0, 1]$ after transformation.
3. Randomly drop textual and parameter value range information in metadata.

## D.4 Inference

At inference time, we choose the decoder input sequence length according to the maximum number of trials to run. E.g. to optimize a function with 18 parameters (highest possible dimension $D$ over our test functions) over 105 trials, we set the input sequence length to be at least $(18 + 3) * 105 = 2205$.

We compute the $(y_{\text{min}}, y_{\text{max}})$ range for function value normalization in the tokenization process with the current minimal and maximum observations. We set $c = 0.2, s = 0.6$ so that all normalized observations fall in the range of $y'_{\text{norm}} \in [0.2, 0.8]$, and the model's $y$ value predicted distribution support, $[0, 1]$, is sufficiently large.

We also use a softmax temperature hyperparameter when predicting function values. We choose the temperature to maximize the log-likelihood of the validation split of each dataset seperately. On RealWorldData, the function prediction temperature is set as 1.1 and on HPO-B it is 1.5. The policy prediction temperature is always set to be 1.

# E Additional experimental results

We provide additional experimental results in this section.

## E.1 Imitating HPO policies

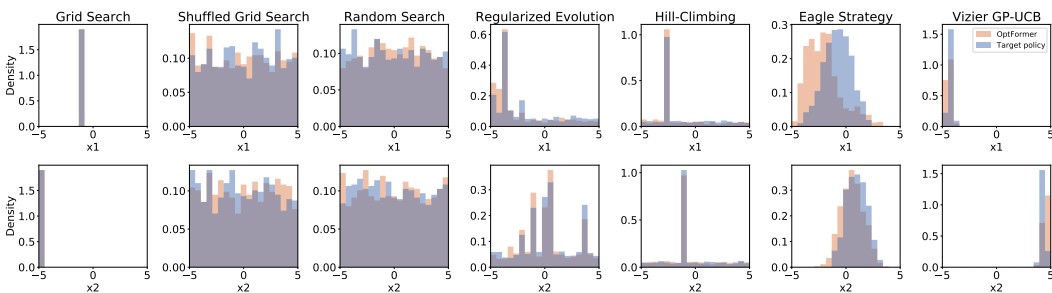

Figure 6: Policy distribution $p(x_{40}^{(d)}|m, \boldsymbol{h}_{39}, \boldsymbol{x}_{40}^{(1:d-1)})$ for $d = 1, 2$ on a 2D GRIEWANK ROSEN-BROCK function.

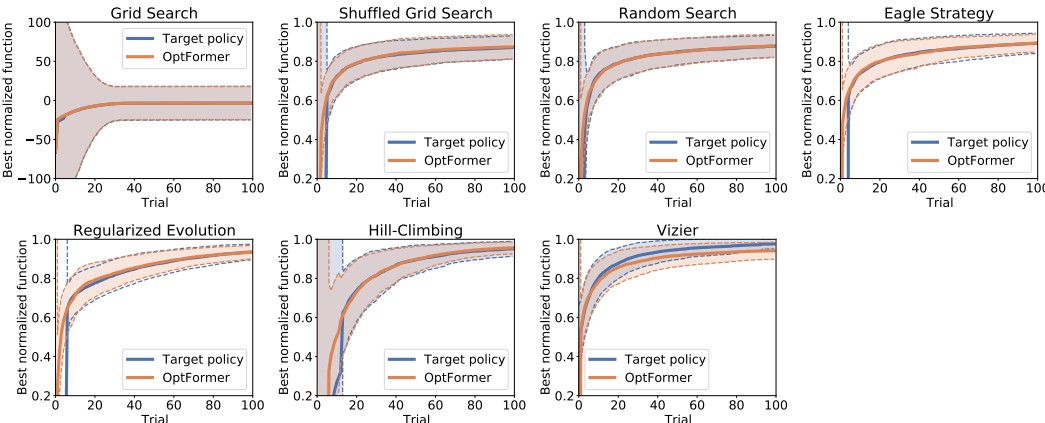

Figure 7: Best normalized function value with std, averaged over 5 test functions each with 100 runs.

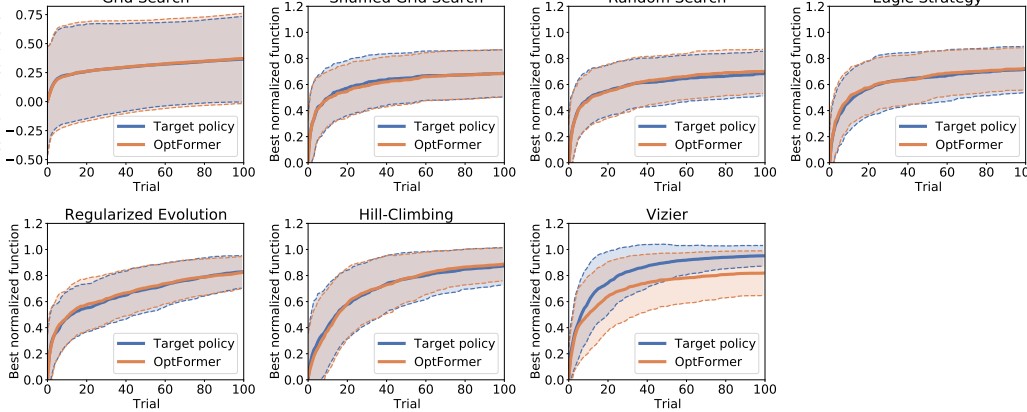

Figure 8: Best normalized function value of LINEAR SLOPE with std, averaged over 100 runs.

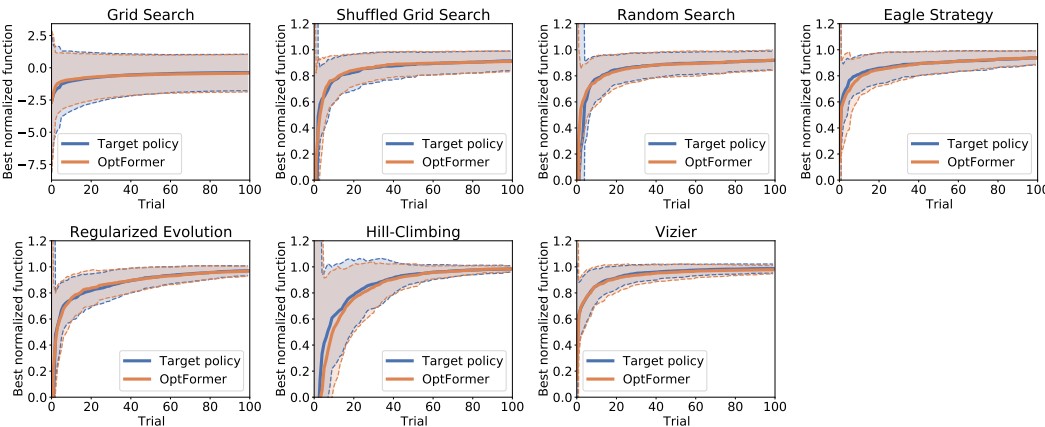

Figure 9: Best normalized function value of ROSENBROCK ROTATED with std, averaged over 100 runs.

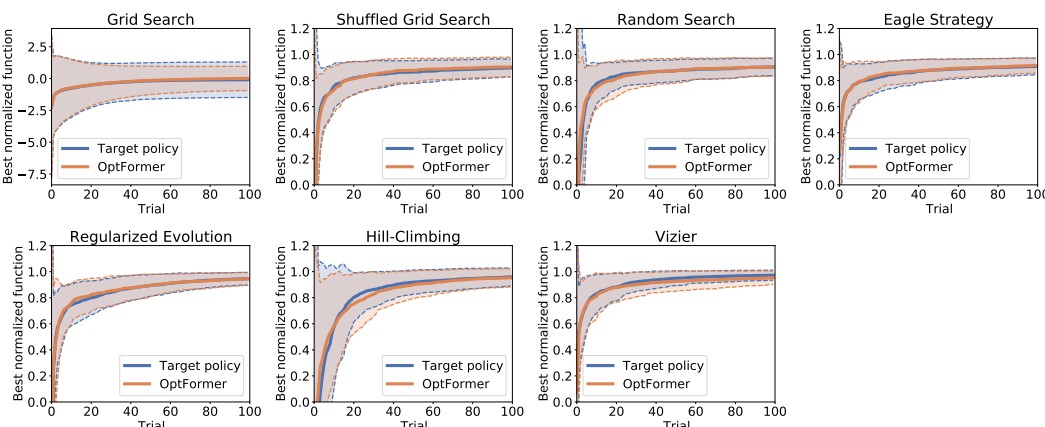

Figure 10: Best normalized function value of SUM OF POWERS with std, averaged over 100 runs.

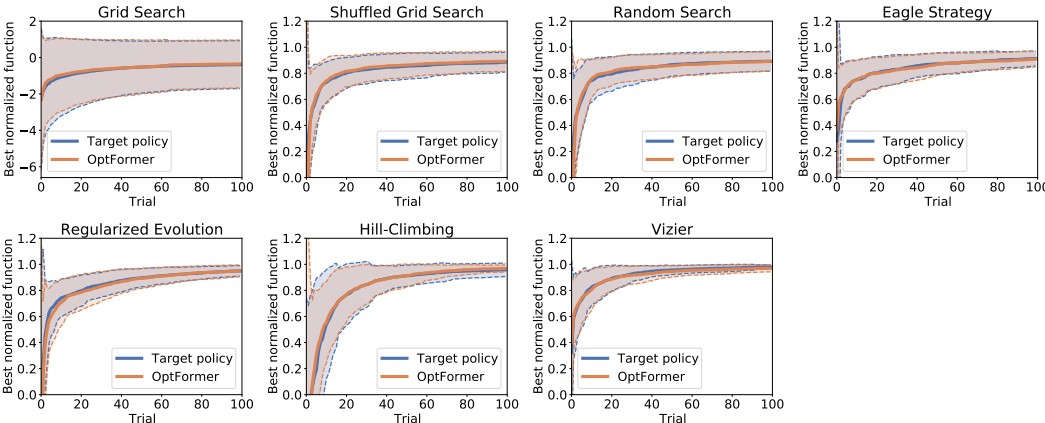

Figure 11: Best normalized function value of GRIEWANK ROSENBROCK with std, averaged over 100 runs.

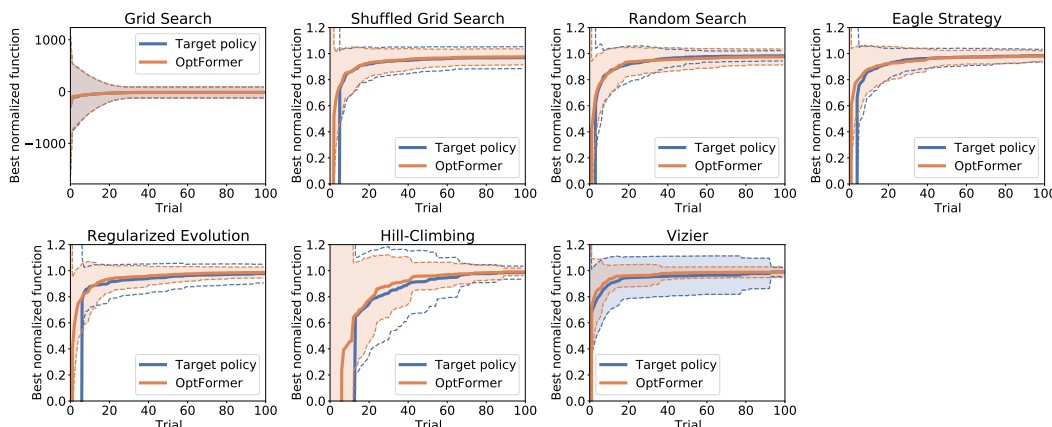

Figure 12: Best normalized function value for LUNACEK with std, averaged over 100 runs.

## E.2 Learning priors for hyperparameter response functions

We apply the same goodness-of-fit analysis on function prediction from Section 6.2 to the test split of HPO-B. The results are shown in Fig. 13.

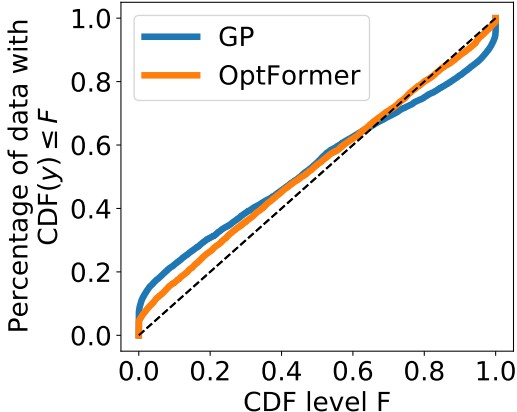

Figure 13: Fitness of predicted CDF(y) on HPO-B test set.

The ECE metric is defined for a classification model. To obtain a similar measurement for a continuous regression model, we convert the continuous regression problem into a multi-class classification problem by discretizing the range $[y_{\min}, y_{\max}]$ for each study into 100 equal intervals. Then, we follow the definition of ECE in [27] and estimate the metric using 10 confidence bins.

## E.3 Augmenting a prior policy with function prediction

**Transfer learning results on HPO-B** Fig. 4 shows the best normalized function values observed so far at each trial. Though HyperBO uses a smaller NN for feature extraction, HyperBO has a flexible mean function, which captures important information that benefits BO in beginning trials. While we implemented a bare-bone FSBO, its performance is still better than ABLR in part thanks to FSBO's use of a squared exponential kernel instead of a dot-product one. Compared to a dot-product kernel with a finite feature space, a squared exponential kernel introduces infinite features.

In Fig. 14 and Fig. 15, we show the performance profiles of all compared methods over 2 different metrics: outperforming 90% of the best function value obtained by all methods at the 50th iteration, and outperforming the median of the best function values obtained by each method at the 50th iteration.

Performance profiling is a performance evaluation tool to compare optimization methods, which is widely used in optimization [69]. In our case, the y-axis is the fraction of tasks that each method succeeds in at different BO iterations (x-axis). The criteria of success depends on the problem itself, and we present performance profiles based on 2 different metrics: outperforming 90% of the best function value obtained by all methods at the 50th iteration, and outperforming the median of the best function values obtained by each method at the 50th iteration.

Despite the relatively better performance of HyperBO, FSBO, and ABLR especially during earlier trials as shown by Fig. 4, these methods do not achieve a high percentage success rate on the 86 HPO-B test functions as reflected by Fig. 15. As pointed out by Wang et al. [58], ABLR, FSBO can be viewed as special cases of HyperBO with specific settings of kernel and mean functions. These methods have guarantees only if each function (corresponding to each task) is an i.i.d. sample from the same GP. However, for some search spaces in HPO-B, there exist surrogate functions that return constant values. The constant surrogate function is unlikely to be an i.i.d. sample from the same GP as other surrogates in the same search space. This means ABLR, FSBO, and HyperBO can be sensitive to how the data is generated and outliers in the training data.

Summarizing the results in Fig. 4, Fig. 14 and Fig. 15, HyperBO is able to achieve very good overall performance on a subset of all search spaces, which leads to a better averaged best normalized function values. It is likely that these search spaces have surrogate functions that meet the i.i.d function sample assumption from Wang et al. [58]. However, if we only look at the fraction of tasks

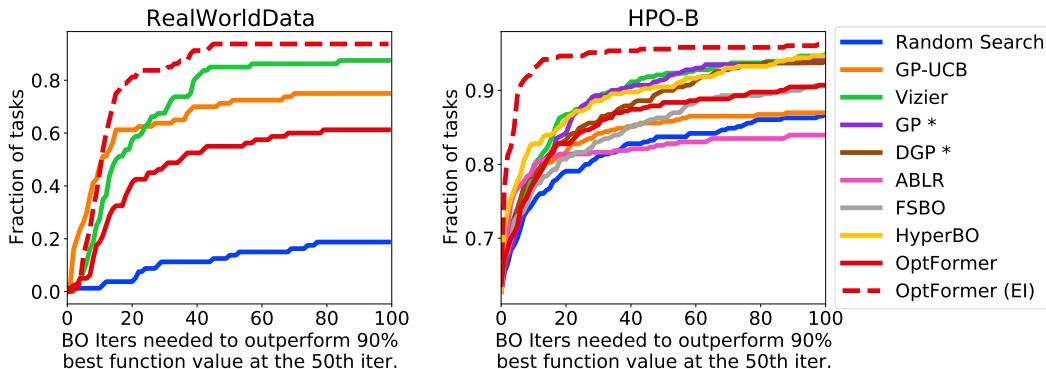

Figure 14: Performance profile on RealWorldData and HPO-B test functions with success threshold: 90% best function value at 50th iteration.

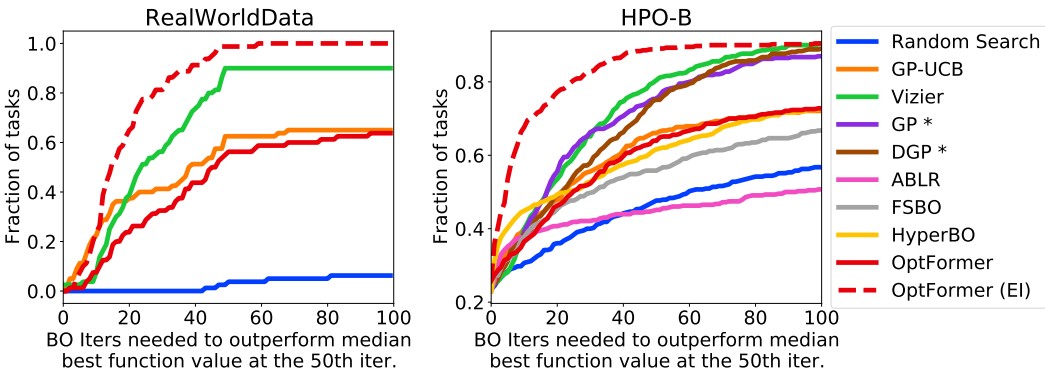

Figure 15: Performance profile on RealWorldData and HPO-B test functions with success threshold: median best function value at 50th iteration.

each method surpasses a success metric, HyperBO may not be a method with superior performance that is comparable to the OPTFORMER. This reveals another benefit of the OPTFORMER: robustness to function outliers.

**HPO-B plotting** We further compare the augmented policies from Section 6.3 to the provided baselines for HPO-B in [5], using the same plotting format from [5] for fair comparison.

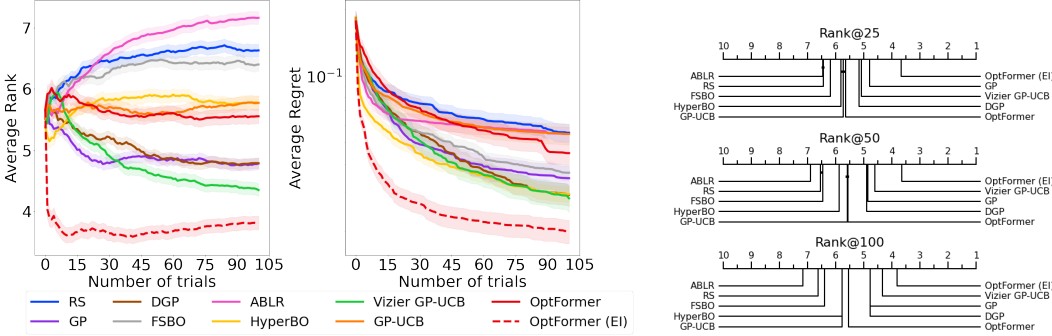

Figure 16: (Lower is better) Aggregated comparisons of normalized regret and mean ranks across all search spaces on the continuous search spaces of HPO-B-v3.

### E.4 Ablation on acquisition functions

We provide additional ablations on acquisition function choices on both the RealWorldData and HPO-B datasets.

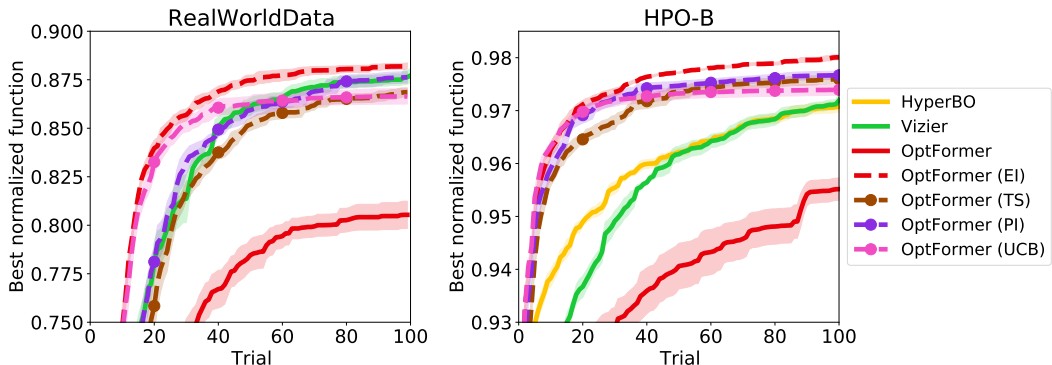

Figure 17: Ablation on the choice of acquisition functions. The plot shows the best normalized function values averaged over HPO-B test functions. Ablation curves are shown with ◯ markers.

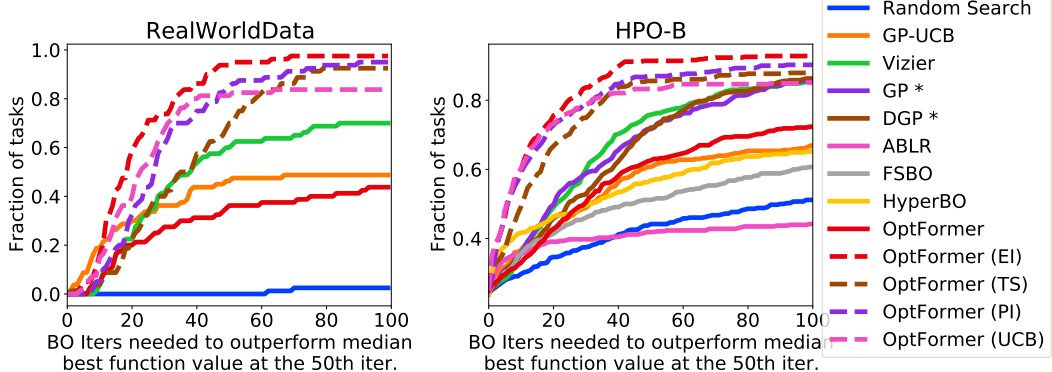

Figure 18: Ablation on the choice of acquisition functions. The plot shows the performance profile metric with success threshold: median best function value at 50th iteration.

In Fig. 17, we compare the Expected Improvement (EI) used in the main body with Thompson Sampling (TS), Probability of Improvement (PI), and Upper Confidence Bound (UCB) with a confidence level of 0.9. We also include the best performing standalone baseline, Vizier, and transfer learning baseline, HyperBO, for reference. We observe that the prior policy is improved by all the acquisition functions. Particularly, OPTFORMER (EI) is the best among all acquisition functions and clearly outperforms all the baseline methods (HyperBO and Vizier) on both datasets across all trial steps. OPTFORMER (UCB) finds good parameter settings as quickly as EI initially, but then becomes saturated early, suggesting a less exploratory behavior than EI. The performance of PI and TS increases more slowly, but keeps improving compared to UCB.

To further bolster this hypothesis, we also compare using performance profiles. As this metric depends on the set of methods being compared, we include all baselines from the main body. As we can see, Fig. 18 demonstrates that augmented OPTFORMER policies, especially OPTFORMER (EI), produce superior performance compared to other baselines.

## E.5 Out-of-Distribution functions

Fig. 19 compares the optimization trajectories of the prior policy OPTFORMER, augmented policies with EI (OPTFORMER (EI)) and Thompson Sampling (OPTFORMER (TS)), against Vizier and Random Search on 5 hold-out test function families from the BBOB benchmark. This assesses their performance on a few commonly used test functions for general black-box optimization. Both variants of the augmented policy obtain comparable or better performance than Vizier on most test functions except OPTFORMER (TS) on the family of Linear Slope functions.

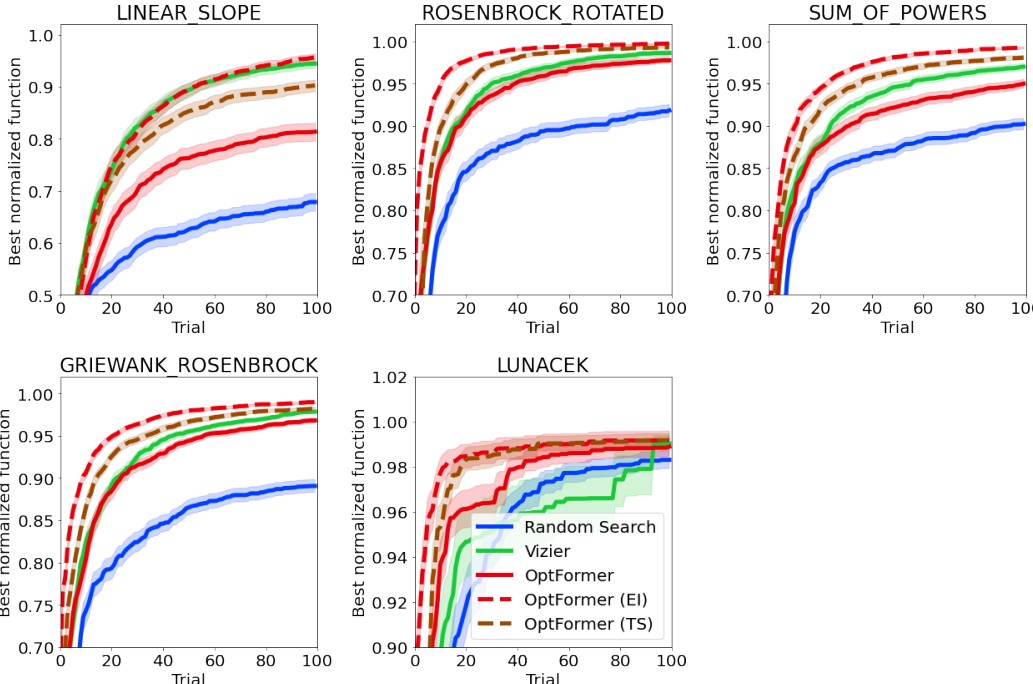

Figure 19: Best normalized function value of a test function in BBOB averaged over 100 runs with std of the mean estimate.

In Fig. 20 and Fig. 21, we further ablate the OptFormer on two machine learning tuning tasks: neural architecture search via NASBench-201 [70] and tuning the learning rate schedule hyperparameters over a live CIFAR-10 training setup using a ResNet-50 from the init2winit benchmark [1]. This assesses their performance on out-of-domain machine learning HPO tasks from the training datasets. Again, OPTFORMER (EI) and OPTFORMER (TS) perform comparably or even better than Vizier. This demonstrate their robust generalization performance over unseen tasks.

[1] https://github.com/google/init2winit

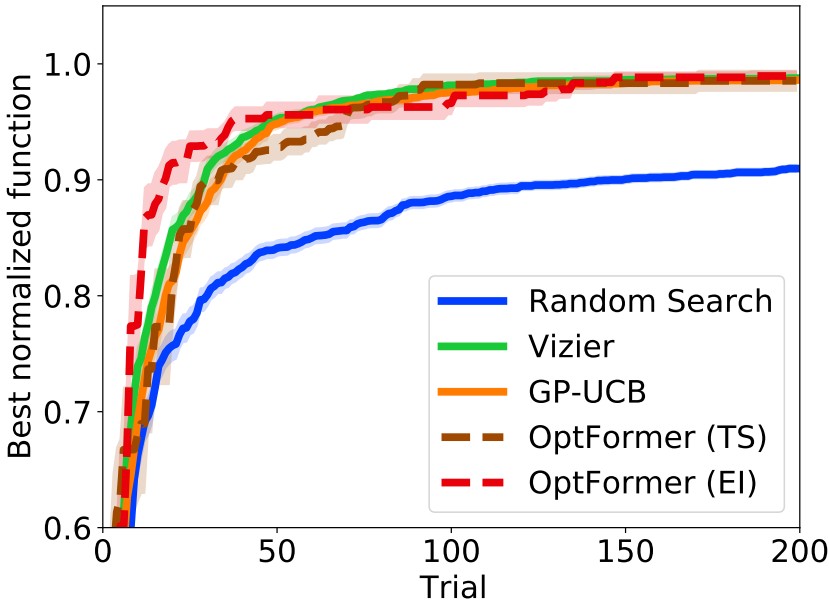

Figure 20: Best normalized function value of NASBench averaged over 10 runs with std of the mean estimate.

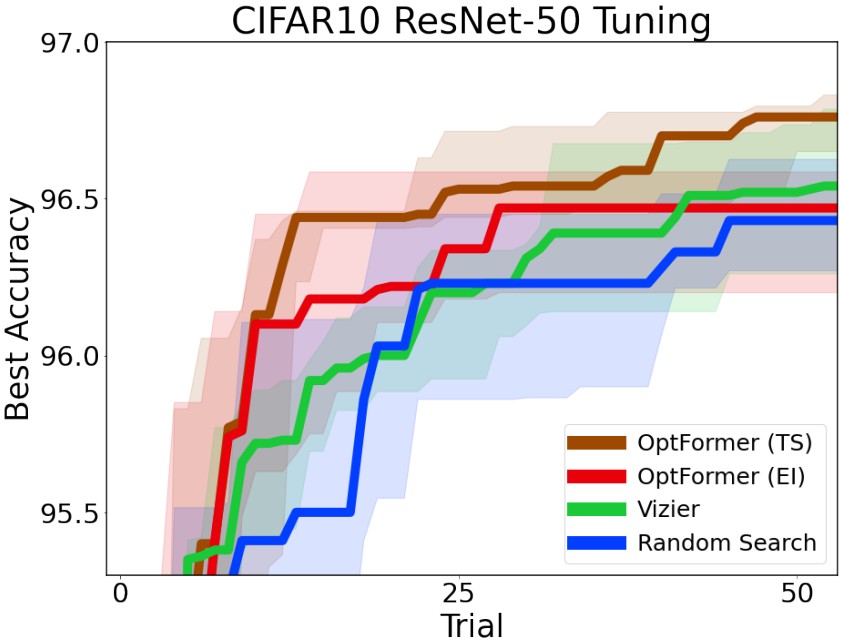

Figure 21: Best CIFAR10 validation accuracy averaged over 10 runs with 25/50/75th percentiles shown.