# OpenReview forum: "Towards Learning Universal Hyperparameter Optimizers with Transformers"
_NeurIPS.cc/2022/Conference — NeurIPS 2022 Accept_

### Official Review · Reviewer_mbxf · 2022-07-08

**Rating:** 6
**Confidence:** 5
**Soundness:** 3 good
**Presentation:** 4 excellent
**Contribution:** 3 good

**Summary:**

The authors train a single Transformer language model (LM) on a variety of tokenized HPO trajectories from a variety of tasks and HPO algorithms. For each HPO trajectory the LM is primed on the task/algorithm (name, search space, metric, algorithm) and is then trained to predict hyper-parameters as well as the optimized metric step-by-step for each trial in order.

**Questions:**

- Is the discretization of scalars for the input important? How would a model perform, where these numbers are normalized in some way and fed to the network directly?
- Do you have an explanation for the strong performance of using RealWorldData for training (Fig. 5 a) compared to the larger HPO-B dataset, when comparing on HPO-B? (Transfer from a smaller dataset is better than in-domain training. This is unusual.)
- What do you mean by "temporal train/test splits" in line 269?
- How do you calculate the Thompson Sampling utility function?

**Limitations:**

The listed limitations are fair, even though one limitation, I would expect there to be, is missing: Handling much longer sequences, as the Transformer is trained with a maximum sequence length.

**Strengths And Weaknesses:**

Strengths
- The results are strong: Reproducing the performance of most algorithms up to 100 trials as well as improving upon them. This could be the path forward for the HPO community.
- The authors provide insightful ablations.
- The method is clearly described and simple.

Weaknesses
- This paper does neither open-source its codebase (built open an open-source codebase), nor the trained model and not even the training data (built upon open datasets). Actually, even a performance number for optimization for follow-up work to compare with is missing, as scores are calculated using an undisclosed metric and the main results are only reported in the form of plots and not tables.
- For RealWorldData (and HPO-B actually, even though less interesting there) a SOTA BO method would be an interesting baseline to add, like HEBO.
- Is the conclusion of Meta-data ablation (line 287) based on a model trained with meta-data? In that case, I would guess the worse performance stems from a train/test distribution shift, rather than from missing metadata really.


Summary:
The results are strong with some evaluation problems. It is not reproducible, though, which in this case is a particularly big problem, as this paper proposes a very new direction for a field of research in which the expertise to reproduce the results based only on descriptions (previous methods require a very different background) and the resources to reproduce the results without data (running HPO with different optimizers on millions of problems) are missing.

---

> ### Author Response · Authors · 2022-08-02
> **Response to Reviewer mbxf**
>
> We are very happy that the reviewer considers our experimental results strong and ablations insightful, and that our work proposes a very new direction! We respond below to the reviewer’s concerns and questions. If our responses have satisfied your concerns and you believe our paper is appropriate for publication, we kindly and respectfully ask our rating to be reconsidered. Thank you very much!
> * Open-sourcing.
>     * Please also see our comment in the general response section. Due to privacy and proprietary concerns, it is prohibited (both ethically and legally) to release our internal RealWorldData dataset, and the privacy of our user data can be compromised from releasing the corresponding trained model. However, **we have released the core code in** (https://github.com/neurips2022optformer/optformer_neurips2022) for review and will add any additional implementation details upon request.
>
>     * In the paper, we show figures instead of tables of numbers because it is visually easier to tell the difference. We will provide all numbers for future comparison in the final revision.
>
> * Comparison with SOTA methods in RealWorldData and HPO-B
>     * We have provided comparisons to multiple competitive standalone (Vizier) and transfer-learning HPO solvers (ABLR, FSBO, HyperBO) in experiments. We will work on more baseline comparisons but we think the current set of experiments should suffice to support our claims in the introduction.
>
> * Meta-data ablation
>     * As described in Appendix D.3, the data augmentation includes randomly removing all metadata in a training example. Thus the model does indeed see examples with and without metadata during training. Therefore, it is not sufficient to explain the worse performance only from a train/test distribution shift.
>
> ## Questions
> * Discretization of scalars
>     * Discretization allows DOUBLE and INTEGER parameters to be treated in the same way as metadata and other types of parameters in the unified serialization scheme. A reasonable alternative is to feed the parameter as a continuous input variable, but it is important to use a discretized value for the output target to learn a flexible discrete distribution. Using L2 loss to regress continuous parameters will suffer from learning multi-modal parameter distributions, which can exist from studies generated from e.g. BO algorithms.
>
> * Strong performance of using RealWorldData for training (Fig. 5 a) …, when comparing on HPO-B? Transfer from a smaller dataset is better than in-domain training
>     * We believe this might be a misunderstanding. In Fig. 5a, the model trained on all data (OptFormer (TS)) is slightly better than the model trained on in-domain HPO-B (OptFormer-H (TS)) which is again better than the model trained on out-domain HPO data RealWorldData (OptFormer-R (TS)), and out-domain black-box data BBOB (OptFormer-B (TS)). This is consistent with the conclusion that both (1) more diverse training data and (2) more relevant data help the model. We will clarify this point.
>
> * "temporal train/test splits" in line 269?
>     * All of the training tuning studies were generated before Feb 2020, and the test studies were generated afterwards up to March 2022. We have included the details in section 6 and appendix C.1 in the revision.
>
> * Thompson Sampling utility function
>     * Inspired by the Thompson sampling method in bandit problems, we define our utility function as a sampled function value $y_i$ at a given location $x_i$ from the predictive distribution. It approximates the search for the maximum location in one function realization on a small sampled index subset ${x_i}$. More formally, $x^{*} = \arg \max_{x_i} y_i$, with $y_i \sim p(y|x_i, …)$ and $x_i \sim \pi_{prior}(x|m,h)$ for all $i$.
>     * We have included the formulation of all the three acquisition functions in Section 4.3 of the revision.
>
> * Limitation on the sequence length
>     * Thanks for the suggestion. We have included the discussion in section 7 of the revision. More scalable architectures such as Performers [1] would help address that limitation.
>
> **References** \
> [1] Rethinking Attention with Performers. ICLR 2021.

---

> > ### Comment · Reviewer_mbxf · 2022-08-04
> > **Reply**
> >
> > Thanks for the detailed reply.
> >
> > Open Sourcing
> > Thanks for open-sourcing more of your code. I think this is still not enough, though, and not all you can do. Especially for follow-up work. Still easily possible without uploading large datasets or compromising use data:
> > 1. Upload a trained model on public data only, e.g. OptFormer-H (TS).
> > 2. Add numbers on public data to the paper in a table in the appendix for with a public normalization function.
> >
> > Meta-data ablation
> > Good point. I overlooked that at the time.
> >
> > Thanks for all the answers to my questions. Here are my answers to some, numbered by the question order.
> >
> > 2. Thank you for clarifying this. I misunderstood this.
> >
> > 4. This is very interesting. I think this is a different (but interesting) acquisition function, I do not think it is Thompson Sampling. At least not what is meant by this in the BO literature. In Thompson Sampling we sample from the distributions over maxima of functions. Let us for example consider a prior that has some fixed noise on all positions eps \in Uniform([0,1]) and consists of simple piece-wise linear functions of the form `[(0,eps),(0.5,0.1+eps),(0,eps)]`. Thompson sampling in the first step would always return 0.5 as this always is the function maximum, but your acquisition function (if I understood it right) does with high probability return something other than 0.5. This is because there is a large likelihood that eps > 0.1 + eps' (for eps and eps' from Uniform([0,1])) but the probability of eps > 0.1 + eps is 0. Do you know what acquisition function that is that you have? I think this would need clarification and motivation in the paper. A good reference for Thompson sampling is the BO book, chapter 7.9. (https://bayesoptbook.com) I think you could theoretically fix this in your setup by conditioning $y_i$ on all $y_j$ with j<i.

---

> > > ### Author Response · Authors · 2022-08-04
> > > **Additional Reply to Reviewer mbxf**
> > >
> > > Thank you for the detailed response.
> > >
> > > Open-Sourcing
> > > * We have also added the checkpoints for OptFormer-H and OptFormer-B at (https://drive.google.com/drive/folders/1iNtdCj66TbzNeQzFMFgbzKyqZIZiP_ex?usp=sharing), and will continue to improve on open-sourcing the code by the camera-ready date for making the OptFormer more easily runnable. However, we hope that our current best efforts are satisfactory, given the time limit and technical challenges of open-sourcing proprietary code with large internal dependencies.
> > >
> > > * For the scoring found in the main paper, we refer the reviewer to the normalization ranges and scheme from the HPO-B paper [1] which we used for the HPO-B results. For transparency, we uploaded all of the results used for normalization and plotting as JSON files in the anonymous drive (https://drive.google.com/drive/folders/1A-B1IW7ZxmGbjNn6tHUknkMH5bP2va82?usp=sharing), and will add all of the extensive numbers including BBOB scores to the Appendix paper.
> > >
> > > For Question 3:
> > > * You are right that our TS acquisition is different from Thompson sampling in the literature because here, we ignore the correlation of sampled functions across inputs. The effect of our acquisition function is to exploit the posterior mean and use the posterior variance for exploration (combined with input sampling from the prior policy). We will clarify this in the revision.
> > >
> > > * Conditioning yi on all sampled yj with j<i will fix the discrepancy and it would be a good idea to test. Nonetheless, now that our model has to predict yi on multiple fascinated yi’s rather than observed real values, it is different from the training setting and may introduce additional modeling errors.
> > >
> > > * Given the overall better performance EI over other acquisition functions in our ablation study (Figure 6 in Appendix E.1, all variants are close), we plan to use the more standard EI in place of TS in the main text and include TS together with PI, UCB in ablations. We’ve repeated the main body’s TS ablation studies in Figure 5 using EI and the same conclusion remains.
> > >
> > > **References** \
> > > [1] HPO-B: A Large-Scale Reproducible Benchmark for Black-Box HPO based on OpenML (https://arxiv.org/abs/2106.06257)

---

> > > > ### Comment · Reviewer_mbxf · 2022-08-05
> > > > **Reply**
> > > >
> > > > Very nice! I upgraded my rating, since these are significant improvements. This is under the assumption that these links will work indefinitely and are included in the paper such that follow up work is much easier. Thank You!

---

### Official Review · Reviewer_yjnm · 2022-07-13

**Rating:** 7
**Confidence:** 4
**Soundness:** 3 good
**Presentation:** 4 excellent
**Contribution:** 4 excellent

**Summary:**

The paper studies hyperparameter transfer with metadata in an "open-set" setting -- allowing different configuration spaces across tasks. A transformer-based hyperparameter tuner, namely OptFromer, is proposed to predict policy and response function values (e.g., validation performance) in a sequence-to-sequence training style, where the learned policy maps text-based metadata to pre-discretized hyperparameter configurations. To my best knowledge, the proposed method is the first HPO method that learns prior knowledge from the collected text-based configurations. Experimental results on one collected real-world dataset and two public benchmarks were provided in terms of the policy behavior imitation, response function prediction, and HPO.

**Questions:**

- Table 4 shows more calibrated results of OptFormer than the GP-based methods. It is well known that GP could provide well-calibrated uncertainty estimates. Yet, in accordance with Eq (3-5), it seems like the proposed method just follows the standard supervised training strategy. Any insights into why OptFormer could show a better calibration result? Is the reason due to using the transformer architectures [40]? I'm also curious about the ECE comparison between OptFormer and OptFormer (TS)
- As shown in Fig. 4, the Vizier performs slightly better than OptFormer (TS) on the RealWoldData dataset. While the paper implies the reason as the GP surrogate-based test functions, it is unclear why OptFormer performs much better than GP-UCB. Also, the comparison results between RealWorldData (*mixed algorithms*) and HPO-B (*controlled algorithms*) cast a shadow on the biased issue of the proposed OptFormer.

**Post after rebuttal**
Thanks for providing a detailed response to the questions. Due to my travel schedule, I haven't got a chance to further discuss with the authors. Yet, most of my previous concerns were well addressed. Particularly, it would be interesting to add an LSTM baseline in future work and explore more the calibration of HPO from a pre-training perspective. I would like to champion this work by upgrading my score.

**Strengths And Weaknesses:**

Pros:
- The proposed hyperparameter prior learning method is orthogonal to existing GP-based hyperparameter transfer methods in two folds: 1) it relaxes the limitation of sharing the same configuration space across different tasks, and 2) the data-driven approach given by training a transformer model significantly improves the efficiency.
- It is interesting and novel to learn prior knowledge from text-based metadata. The seq-2-seq supervised learning framework is technically sounded with proper practical treatments. Moreover, the transformer structure is well-motivated to capture both symbolic and numerical manipulation.
- While some necessary implementation details are missing in the manuscript, the augmented HPO policy with Thompson Sampling provides a good implementation similar to offline RL.
- Extensive experimental results demonstrate the effectiveness of the learned HPO policy on its well-calibrated predictions and utility performance.

Cons:
- The main concern of this work is its unclear **meta knowledge** learned from text-based metadata. How does the proposed OptFormer indeed imitate the other HPO algorithms? Does the transformer simply memorize the choices given by different HPO algorithms? Can the proposed method adapt to more complex algorithms (e.g., hypergradient-based) and large-scale hyperparameters?
- The learned prior seems risky to be biased on closed-set model architectures and tasks (datasets). It remains unclear if the HPO policy can be generalized to **unseen** tasks. I may miss something in the appendix; yet, it will be helpful to give more details about the split of training/test set in the RealWorldData. A non-overlapping split over the tasks or algorithms will be more convincing.
- One major technical contribution of this work is to introduce transformers for learning HPO priors. Hence, it is expected to give an ablation study regarding this architecture choice. One baseline based on RNN (e.g., LSTMs or GRUs) will be useful to validate this point empirically.

---

> ### Author Response · Authors · 2022-08-02
> **Response to Reviewer yjnm**
>
> Thank you for your encouraging comments! We are glad you find our work interesting and novel, and as well as provides an orthogonal approach to existing GP-based hyperparameter transfer methods. Please see our answers below to your concerns and questions.
>
> * Unclear meta knowledge
>     * We agree that it is hard to diagnose the internal representation of the learned meta knowledge of the Transformer model. However, we can understand the model’s behavior by assessing its output policy. We suspect the model does more than memorize the choices given by training algorithms but learns the internal algorithm logic because we test on holdout functions in Section 6.1 that have different characteristics from training functions (full list of functions is given in Appendix C.1). With sufficient training data and large enough architecture, it should be possible to learn more complex algorithms on larger search dimensions.
> * Biased on closed-set model architectures and tasks (datasets)
>     * That’s a very good question. The transfer learning approach allows one to perform better over test tasks similar to training tasks, but potentially in turn worse in other tasks. However, we provided a broad range of experiments to evaluate the model’s generalization performance in this paper, including the new experiments (Appendix E.5-6) over unseen BBOB functions, NASBENCH-201, and live CIFAR10 learning rate tuning.
>
>     * We provided the details of the RealWorldData dataset’s collection in Appendix C.1. We split the dataset in temporal order to avoid information leak. In addition, the users who generated the test tuning experiments only started to use the tuning service *after* all the training studies were generated. We believe those test functions consist of a presentative set of functions for machine learning hyperparameter tuning tasks and non-overlapping with the training functions.
>
> * Ablation study regarding this architecture choice
>     * This is a good suggestion. The main motivation of this paper is to provide a universal interface for HPO that can be modeled by a sequence model. The Transformer is a natural choice over RNN architectures such as LSTM due to the enormous success of transformers in all areas of ML in recent years. We hypothesize already that an LSTM will struggle to learn due to the long sequence length of our problem (e.g. 1000 tokens from 100 steps * 10 parameter dimensions) as e.g. [1] applied an LSTM for HPO for fixed input dimensions but found it hard to scale to even a sequence length of 100 without using curriculum learning. That being said, it would still be useful to include an LSTM baseline in our experiments.
>
>
> ## Questions
> * More calibrated results of OptFormer than the GP. ECE comparison between OptFormer and OptFormer (TS)
>     * We conjecture OptFormer learns better calibrated results because of the transfer learning setup. As shown in the relationship below, the set of HPO tasks is a subset of all possible blackbox optimization functions. We expect OptFormer learns a better and more specific prior from HPO data than a GP model, which possesses a very general prior assumption on local smoothness which could work broadly for blackbox optimization but maybe not as well over a specific subset of tasks.
>         * $\text{Training HPO tasks} \approx \text{Test HPO tasks} \subset \text{HPO tasks} \subset \text{Black-box optimization tasks}$
>
>     * Furthermore, OptFormer (TS) is a policy to propose $x$. It uses the learned function prediction ability to rank samples from the prior policy and does not modify its predictive distribution of $y$ given $x$. Thus we cannot compare ECE between OptFormer and OptFormer (TS).
> * Vizier performs slightly better than OptFormer (TS) on the RealWoldData dataset. OptFormer performs much better than GP-UCB.
>     * As shown in an ablation study in Appendix E.1, Figure 6, OptFormer (EI) performs even better than Vizier on RealWorldData. Vizier combines GP-UCB with a trust-region method that works more robustly in practice than a vanilla GP-UCB. However, tuning the hyperparameters of GP-UCB to work broadly is tricky in practice. We suspect this is another reason GP-UCB underperforms against Vizier and OptFormer (TS) / (EI). Vizier and GP-UCB hyperparameters are explained in Appendix B.1.
>
> **References** \
> [1] Learning to learn without gradient descent by gradient descent. (ICML 2017)

---

### Official Review · Reviewer_MJPS · 2022-07-13

**Rating:** 7
**Confidence:** 4
**Soundness:** 3 good
**Presentation:** 3 good
**Contribution:** 3 good

**Summary:**

The paper describes a new meta-learning approach for hyperparameter optimization (HPO) based on a transformer model. The model is trained on offline generated data that includes the metadata that characterizes the optimization problem, for example the search space and  and the history of observed trials, i.e function values and input configuration. During inference time, the model can be combined with HPO policies, such as Thompson sampling or upper confidence bounds to suggest new hyperparameter configurations.

**Questions:**

- Section 6.2: How do you compute the predictive distribution p(y|...)? My understanding is that the transformer only predicts discrete outputs with [0, Q)

- Section 6.4 prior policy: How is Random Search combined with Thompson sampling (Random Search-TS)?

- What was the computational budget to train the transformer model and how long did it train?

- Do you plan to open-source the dataset and the code to reproduce the results?

**Limitations:**

While the method improves across a set of baselines, it does not improve yet over more sophisticated algorithms such as Vizier on real world datasets. I assume that it would also not outperform current state-of-the-art methods that early stop poorly performing configurations, such as Hyperband or BOHB. However, I think this is fine for a research paper, but it would not be sufficient for production.

**Strengths And Weaknesses:**


### Reason for overall rating

Current transfer learning approaches for HPO are limited to a fixed search space and the same underlying machine learning model and only transfer knowledge across different datasets. This paper presents, to the best of my knowledge, the first approach that enables meta-learning across these different dimensions. While I don't think the method is ready for a practice yet, the paper marks a first important step towards more universal HPO methods.


### Strengths

The paper aims to learn a more general meta-learning approach for HPO, that generalizes not only across datasets, but also machine learning methods and search spaces. This, in theory,  allows to access a much large amount of offline data and allows to generalize across different domains.

Overall, I found the different parts of the paper, e.g tokenization, inference and decoding of the model, well motivated and clearly explained.

`The empirical evaluation of the paper contains sensible set of baselines. Also the ablation study provides convincing insights in the proposed approach.



### Weaknesses

It remains a bit unclear how well this model generalizes beyond the training data. For example, what would happen if the method is applied to other problem domains, such as neural architecture search or general gradient-free optimization problems. Similarly, how does the method scale with the dimensionality of the search space?

The dataset generation seems a bit ad-hoc. Is it really necessary to include trajectories of such a large variety of optimizers or would it be sufficient to limit to few state-of-the-art optimizers? This could potentially reduce the dataset size and would allow to us a smaller architecture.

The paper could elaborate on the pre-training of the model. For example, how did different design decision of the network architecture effect downstream performance? How difficult was the pre-training, e.g did you have to restart from previous checkpoints, etc ?

---

> ### Author Response · Authors · 2022-08-02
> **Response to Reviewer MJPS**
>
> We really appreciate your positive comments and constructive feedback. Our intention of this work is to take the first step towards universal HPO methods. Below are our responses to your questions.
>
> * Generalization beyond the training data
>     * The RealWorldData dataset contains very diverse HPO tasks that, in our opinion, lead to good generalization with comparable or better performance than the Vizier algorithm. Please note that test functions from the RealWorldData dataset were collected after the training subset in time and do not intersect in terms of the users who generated those studies. Those should represent a good set of out of domain test functions, as different users tune very different objectives.
>
>     * Upon your request, we have conducted additional experiments for out-of-domain functions, including unseen BBOB functions (Appendix E5), along with NASBENCH-201 and tuning over a CIFAR10 training pipeline (Appendix E6). Please refer to our general response for details. Extensions to more flexible search spaces with conditional dependencies for general combinatorial spaces such as NAS will be in our future work.
>
> * Dataset generation
>     * In our additional experiments, we find that it is sufficient to train OptFormer from trajectories generated by Vizier in the BBOB and HPO-B datasets (RealWorldData is fixed without our control of the algorithm). In our paper, we show that a single model can be trained on all the data and perform multiple tasks, but it is not necessary to generate the data using many optimizers. A single optimizer that generates data well should lead to better performance.
>
> * Pre-training models
>     * We find the default T5 model architecture works well for our problem. We only modify the loss function with weighting so that the model does not need to predict the separating tokens. The only hyperparameter we tuned on a validation set is the dropout rate. The model is trained from scratch.
>
> ## Questions
> * Compute the predictive distribution $p(y|...)$
>     * As explained in the “Function prediction” paragraph in Section 4.3, we construct a piecewise uniform distribution in the continuous interval of [0, Q) and then map linearly back to the original function range.
>
> * Random search + Thompson sampling
>     * By “Thompson sampling” we mean an acquisition function inspired by the Thompson sampling method in bandit problems. We define it as a sampled function value $y_i$ at a given location $x_i$ from the predictive distribution. It approximates the search for the maximum location in one function realization on a small sampled index subset ${x_i}$. In mathematical notation, this corresponds to:
>  $x^{*} = \arg \max_{x_i} y_i$, with $y_i \sim p(y|x_i, …)$ and $x_i \sim \pi_{prior}(x|m,h)$ for all $i$.
>
>     * The difference between OptFormer (TS) and Random Search (TS) is that we sample locations x_i from a uniform distribution instead of the prior policy learned by OptFormer. We have included the formulation of all the three acquisition functions in Section 4.3 of the revision.
>
> * Computation budget for model training
>     * As explained in Appendix D.2, we trained the model on a 4x4 TPU-v3 slice (16 devices in total) for up to 1M steps. We performed early stopping once the model started to overfit after about 200K steps. It takes about 1.4 days to train for 200K steps.
>
> * Open-source
>     * We recently submitted our code at  https://github.com/neurips2022optformer/optformer_neurips2022/. We will also work on open-sourcing the terabyte-sized datasets created from public benchmarks. Please also see our comment in the general response section, thank you!

---

### Official Review · Reviewer_C6k8 · 2022-07-17

**Rating:** 4
**Confidence:** 3
**Soundness:** 3 good
**Presentation:** 2 fair
**Contribution:** 2 fair

**Summary:**

The authors propose using Transformer to imitate the hyper-parameter optimization. The authors claim this is the first work in the area, and claim the proposed method exceeds several classic HPO methods.

**Questions:**

1. how do you justify max_i\in{1:t}(y_i-y_rand)/(y_max-y_rand) a good evaluation metric? Why don't you use the y_i and plot as a range?

3. Fig.4 made a few strong claims, especially that OPT-Former performs better than GP-UCB. I'm trying to understand the underly causes. Here is my guess: opt-former is trained on datasets that potentially contain tasks with similar distributions tested in Fig.4. The advantage of GP-UCB is to start without any priors and gradually approximate the underlying function contours by sampling. Without any prior data, I'm surprised that OPT-Former can perform better than GP-UCB. I'd be happy to see if I'm wrong, and it will be compelling if the authors can provide an anonymous link to the code for a quick comparisons. (key factor for me to improve the score)

4. The methods used in this paper does not fully capture the landscape of black box optimization solvers today. The authors may find the following repo to be useful. (feel free use at your discretion, it is just a suggestion)

a. https://botorch.org/

b. https://github.com/facebookresearch/nevergrad

c. https://github.com/facebookresearch/LaMCTS

d. https://github.com/uber-research/TuRBO
These repos have encapsulate several exciting BBO algorithms today.


**Limitations:**

no limitation found

**Strengths And Weaknesses:**

Strength:
1. very comprehensive evaluations.
2. seemly reasonable idea.

Weakness:
1. limited novelty: perhaps applying transformer to HPO could be counted as a novel point, but I feel this is not enough.
2. The paper is difficult to read, please try to improve the readability. Many places in the paper tries to impress the readers with sophisticated terms even on very simple concepts.

---

> ### Author Response · Authors · 2022-08-02
> **Response to Reviewer C6k8**
>
> We thank the reviewer for their comments. Please see our answers to concerns and questions below.
>
> * Limited novelty
>     * We definitely appreciate that applying Transformers to HPO is considered a novel point. However, please note that this is also the first ever method that makes it possible to meta-learn a single HPO model from scratch on a large hyperparameter tuning dataset. In particular, it is very nontrivial to develop a serialization scheme for unifying tuning tasks over different search spaces and contents.
> This is also pointed out by other reviewers:
>         * Reviewer **MJPS**: “This paper presents, to the best of my knowledge, the first approach that enables meta-learning across these different dimensions. While I don't think the method is ready for a practice yet, the paper marks a first important step towards more universal HPO methods.”
>         * Reviewer **yjnm**: “It is interesting and novel to learn prior knowledge from text-based metadata.”
> * Readability
>     * Thanks for pointing this out. Could you please provide more specific examples to inform us on our use of “sophisticated terms” to help us improve the paper?
>
> ## Questions
> * Normalization in the evaluation metric.
>     * It is necessary to normalize function values in order to provide an aggregated metric across test functions (500 in BBOB, 16 in RealWorldData and 84 in HPO-B) of different orders of magnitude in scale. Otherwise, the performance on test functions with the largest range will dominate the metric. It is a very common practice in various literature; see e.g. [1, 2, 3, 4, 5, 6]. We have clarified it in section 6 of the revision.
> * OptFormer comparison with GP-UCB
>     * Please note that all models require priors, learned or hand-tuned, to perform inference. Often, users manually tune the prior of GP models to inject prior knowledge on the local smoothness of the objective landscape by tuning associated hyperparameters (choice of kernel, length scale, variance, etc). In contrast, OptFormer takes the meta/transfer-learning approach and learns implicit priors from HPO dataset. We do not claim OptFormer will be better than GP-UCB over all possible black-box optimization tasks, but do claim that OptFormer can learn a better prior for HPO tasks than GPs. Please take the following relationship as a reference:
>         * $\text{Training HPO tasks} \approx \text{Test HPO tasks} \subset \text{HPO tasks} \subset \text{Black-box optimization tasks}$
>
> * Open-Sourcing
>     * Please also see our general comment in a separate reply. We have released the core code anonymously at https://github.com/neurips2022optformer/optformer_neurips2022. Model training is based on an already open-sourced T5X codebase (https://github.com/google-research/t5x) as explained in Appendix D.2.
> * Full landscape of black box optimization solvers
>     * Thanks for the references. In the updated draft, we have appended a comparison between Vizier and BoTorch in Appendix E.7 to demonstrate that Vizier is a competitive algorithm already. Also, in the original draft, we provided comparisons to multiple competitive standalone (Vizier) and transfer-learning HPO solvers (ABLR [7], FSBO [8], HyperBO [9]) in the experiments. We are willing to work on more baseline comparisons but we believe the current set of experiments should suffice to support our claims in the introduction.
>
> **References** \
> [1] HPO-B: A Large-Scale Reproducible Benchmark for Black-Box HPO based on OpenML. (https://arxiv.org/abs/2106.06257) \
> [2] Bayesian Optimization is Superior to Random Search for Machine Learning Hyperparameter Tuning: Analysis of the Black-Box Optimization Challenge 2020 (JMLR 2021) \
> [3] Google Vizier: A Service for Black-Box Optimization (KDD, 2017) \
> [4] HEBO: Pushing The Limits of Sample-Efficient Hyperparameter Optimisation (JAIR 2021) \
> [5] Taskset: A Dataset of Optimization Tasks (arXiv 2021) \
> [6] Frugal Machine Learning (https://arxiv.org/abs/2111.03731) \
> [7] Multiple Adaptive Bayesian Linear Regression for Scalable Bayesian Optimization with Warm Start (NeurIPS 2017) \
> [8] Few-Shot Bayesian Optimization with Deep Kernel Surrogates (ICLR 2021) \
> [9] Pre-trained Gaussian processes for Bayesian optimization (https://arxiv.org/abs/2207.03084, code: https://github.com/google-research/hyperbo)

---

> > ### Author Response · Authors · 2022-08-09
> > **We would appreciate interacting with you**
> >
> > Dear Reviewer,
> >
> > Once again, thank you very much for your valuable time spent on our submission and your thoughtful reviews!
> > As the Author-Reviewer discussion period is coming to an end soon, we wanted to check in with you if we have addressed your questions and concerns, and if we provided all information required for making your final evaluation. We would appreciate interacting with you.
> >
> > Thank you again for your efforts!

---

### Author Response · Authors · 2022-08-02
**General Response to All Reviewers**

We graciously thank all of the reviewers for their time and work in providing feedback on our paper, which will be very useful in improving its quality. We are also very happy that several reviewers consider our work novel and interesting supported by **strong experimental results**, with some comments even considering our work **a potential and important path forward in HPO!** Below, we address common questions and concerns:

## Better results from using an Expected Improvement (EI) acquisition function, and additional out-of-domain experiments

We would like to emphasize that both the TS (main paper) and other variants especially EI (Appendix E.1) of OptFormer obtained comparable or better performance on challenging benchmarks over strong baselines.

Upon the request of reviewer MJPS, we now provide additional experiments (Appendix E.5-6) to compare OptFormer (TS) and (EI) with Vizier on two sets of test functions: (1) hold-out **BBOB** test functions from general black-box optimization (we only compared the imitated policies in Sec 6.1 in the initial submission) (2) out-of-domain HPO tasks: **NASBench-201** [1] for neural architecture search and tuning a **live pipeline for training ResNet-50 on CIFAR-10** [2].

This demonstrates evidence that OptFormer can learn robust underlying representations of Bayesian optimization data acquisition strategies.



## Key Message of our paper
We would like to respectfully emphasize that the key message of this paper is not to obtain a SOTA HPO method, but rather provide a new avenue of research. Our OptFormer is the first work that demonstrates the promise of applying large sequence models to take advantage of offline HPO data, and thus opens the door to many more possibilities over Bayesian Optimization methods.



## Open Sourcing our Code
For this submission we have now anonymously included the code for:
* Tokenization and preprocessing (https://github.com/neurips2022optformer/optformer_neurips2022/tree/main/converters)
* Model + policy inference (https://github.com/neurips2022optformer/optformer_neurips2022/tree/main/t5x)
* Data generation (https://github.com/neurips2022optformer/optformer_neurips2022/tree/main/augmentations)

The rest of the model training is based on the open-sourced T5X codebase [3] as explained in Appendix D.2. Checkpoints for models trained on public datasets can be found in (https://drive.google.com/drive/folders/1iNtdCj66TbzNeQzFMFgbzKyqZIZiP_ex?usp=sharing).

We will continue to work on open-sourcing the code and find ways to practically release the terabyte-sized generated dataset using public benchmarks. However, due to privacy and proprietary concerns, we may not release our internal RealWorldData dataset, as the privacy of our user data could be compromised from releasing the corresponding trained Transformer model [4].

**References** \
[1] NAS-Bench-201: Extending the Scope of Reproducible Neural Architecture Search (ICLR 2020) \
[2] https://github.com/google/init2winit \
[3] https://github.com/google-research/t5x \
[4] Extracting Training Data from Large Language Models (https://arxiv.org/abs/2012.07805)

---

### Meta-Review · Area_Chair_NSF7 · 2022-08-28

**Recommendation:** Accept
**Confidence:** Certain

**Metareview:**

In this work, authors investigate whether Transformers can be used for hyperparameter optimization. The work is interesting and authors outline how they frame the problem and solve practical difficulties. The resulting method is shown to be able to learn to HPO from historical HPO runs and text-based metadata. Some implementation details are missing and I would encourage the authors to add details when possible (taking into account the reviewers' feedback). The empirical evaluation of the paper contains sensible set of baselines and ablation studies. It is also appreciated the authors put extra effort into open sourcing parts of the code.

**Award:**

No

---

### Decision · Program_Chairs · 2022-09-14

Accept